# Inhibition of NMDA receptors through a membrane-to-channel path

Madeleine R. Wilcox[1,4], Aparna Nigam [1,4], Nathan G. Glasgow [1], Chamali Narangoda[2], Matthew B. Phillips[1], Dhilon S. Patel[2], Samaneh Mesbahi-Vasey[2], Andreea L. Turcu[3], Santiago Vázquez [3], Maria G. Kurnikova [2] & Jon W. Johnson [1✉]

*N*-methyl-D-aspartate receptors (NMDARs) are transmembrane proteins that are activated by the neurotransmitter glutamate and are found at most excitatory vertebrate synapses. NMDAR channel blockers, an antagonist class of broad pharmacological and clinical significance, inhibit by occluding the NMDAR ion channel. A vast literature demonstrates that NMDAR channel blockers, including MK-801, phencyclidine, ketamine, and the Alzheimer's disease drug memantine, can bind and unbind only when the NMDAR channel is open. Here we use electrophysiological recordings from transfected tsA201 cells and cultured neurons, NMDAR structural modeling, and custom-synthesized compounds to show that NMDAR channel blockers can enter the channel through two routes: the well-known hydrophilic path from extracellular solution to channel through the open channel gate, and also a hydrophobic path from plasma membrane to channel through a gated fenestration ("membrane-to-channel inhibition" (MCI)). Our demonstration that ligand-gated channels are subject to MCI, as are voltage-gated channels, highlights the broad expression of this inhibitory mechanism.

[1] Department of Neuroscience and Center for Neuroscience, University of Pittsburgh, Pittsburgh, PA, USA. [2] Department of Chemistry, Carnegie Mellon University, Pittsburgh, PA, USA. [3] Laboratori de Química Farmacèutica (Unitat Associada al CSIC), Facultat de Farmàcia i Ciències de l'Alimentació i Institut de Biomedicina (IBUB), Universitat de Barcelona, Av. Joan XXIII, 27-31, 08028 Barcelona, Spain. [4] These authors contributed equally: Madeleine R. Wilcox, Aparna Nigam. ✉email: jjohnson@pitt.edu

N-methyl-D-aspartate receptors (NMDARs) are ionotropic glutamate receptors (iGluRs) present at most excitatory synapses in mammalian brains. Among iGluRs, NMDARs exhibit unique features including voltage-dependent block by $Mg^{2+}$, high permeability to $Ca^{2+}$, and slow deactivation kinetics[1–3]. $Ca^{2+}$ influx through NMDARs activates multiple intracellular signaling pathways involved in synaptic plasticity, and contributes to learning and memory[1,2]. Pathological NMDAR activation, however, is implicated in Alzheimer's disease, schizophrenia, major depressive disorder, and many other neurological and neuropsychiatric disorders[4,5].

NMDARs are heterotetramers typically composed of two GluN1 and two GluN2(A-D) and/or GluN3(A-B) subunits. NMDARs can assemble as diheteromers (e.g., GluN1/2A) or tri-heteromers (e.g., GluN1/2A/2B)[1]. NMDAR inhibition has been extensively studied to understand receptor structure and function, and to develop improved therapeutics. NMDAR channel blockers are inhibitors that block current by binding to the "deep site" at the outer tips of the pore-lining M2 reentrant loops near the middle of the ion channel[6–9]. Channel blockers typically can access their binding site only when the channel is open, and thus are also referred to as open channel blockers. NMDAR channel blockers examined for therapeutic use include $Mg^{2+}$, amantadine, ketamine, MK-801, dextrorphan, phencyclidine (PCP)[10–14], and memantine. Memantine has found particular clinical success as an FDA-approved treatment for Alzheimer's disease[15], and is effective for treatment of numerous other pathological conditions[16,17].

Memantine inhibition has also been observed following exposure of NMDARs to memantine in the absence of agonist, producing "superficial site" or "second site" inhibition[6–8,18,19]. Memantine was found to bind with weak apparent membrane voltage ($V_m$) dependence to a second site that was hypothesized to be superficial to the channel gate and accessible when NMDARs are closed[6–8,18,19] (but see ref. [20]). It initially was assumed that occupation of the second site caused NMDAR inhibition. However, this assumption was demonstrated to be incorrect[8]. Second site inhibition was found to require at least 2 steps: memantine first occupies a second site without inhibiting NMDAR-mediated current, and then transits from the second site to the deep site, where inhibition occurs. This mechanism suggests that "second site inhibition" is a misnomer; occupation of the second site does not produce inhibition, but rather sequesters memantine in a location from which it subsequently can transit to the deep site. The nature and location of the second site has remained a mystery.

Here we test the hypothesis that the plasma membrane is the second site. Previous work suggested the plasma membrane is an important route through which lipophilic molecules can access membrane-associated proteins[21], including voltage-gated $Na^+$ channels (VGSCs)[22–28] and possibly NMDARs[29–33]. Integrating results from NMDAR models, custom-synthesized blockers, and electrophysiological experiments, we conclude that uncharged memantine can enter the membrane and transit to the deep site upon NMDAR activation through gated, membrane-facing fenestrations. Based on our findings we renamed second site inhibition "membrane-to-channel inhibition" (MCI). Our results reveal that one of the most extensively studied classes of clinically significant neuroactive drugs can bind to NMDARs via either of two mechanisms.

## Results

**Quantification of MCI**. To quantify memantine MCI without contamination by "traditional" channel block (mediated by memantine entry into the open channel from the extracellular solution), the protocol shown in Fig. 1a was applied to GluN1/2A

NMDAR-expressing tsA201 cells at $-65$ mV[7,8,18] unless otherwise specified. The protocol consisted of the following steps (Fig. 1a): 1 mM glutamate (Glu) was applied for 20 s and control NMDAR-mediated current before MCI ($I_{Control1}$) was measured; control extracellular solution was applied for 10 s to allow full NMDAR deactivation; memantine in 0 Glu was applied for 30 s; memantine was washed away by a 1 s application of control solution; 1 mM Glu was reapplied for 20–30 s and NMDAR-mediated current reflecting MCI and recovery from MCI ($I_{MCI}$) was measured; control solution was applied for 41 s; 1 mM Glu was reapplied for 20 s and control NMDAR-mediated current after MCI ($I_{Control2}$) was measured. Figure 1a shows the full protocol (left); an overlay of $I_{Control}$ ($I_{Control} = (I_{Control1} + I_{Control2})/2$) and of $I_{MCI}$ (middle); and a point-by-point ratio (Methods) of $I_{MCI}/I_{Control}$ (right). The minimum value of the $I_{MCI}/I_{Control}$ point-by-point ratio was measured (Fig. 1a, right) and normalized to the minimum value of control ratios (Methods) to quantify fractional current during MCI ("Min $I_{MCI}/I_{Control}$").

Memantine was applied for 30 s because longer applications did not increase inhibition[7]. A 1 s wash after memantine application was used because: it is brief enough to allow MCI measurement (time constant of recovery from memantine MCI is ~2 s (ref. [7])); it is long enough to completely eliminate memantine from the extracellular solution (Methods), ensuring that MCI measurements were not contaminated by traditional channel block.

**Memantine MCI depends on extracellular pH**. A key previous finding was that memantine associates with the "second site" equally well at +35 or at $-65$ mV[8]. Based in part on the $V_m$ independence of memantine association with the second site, we proposed that the "second site" might represent a pool or reservoir of memantine in the plasma membrane[8]. Here we test that hypothesis.

Memantine is a primary amine in equilibrium between charged (protonated) and uncharged (unprotonated) forms (Fig. 1b). The charged form of memantine can enter lipid bilayers, where the charged nitrogen contacts lipid headgroups[34]. Uncharged memantine is highly hydrophobic[27,35–37] and thus resides predominantly in the membrane.

At physiological pH memantine is predominantly in the charged form ($pK_a = 10.4$; ref. [36]). The fraction of uncharged memantine in aqueous solution increases as pH increases (Fig. 1b). If second site occupation reflects uncharged memantine in the membrane, then increasing extracellular pH should increase the uncharged memantine concentration ([uncharged memantine]) both in solution and in the membrane, increasing MCI. We tested this prediction by measuring the dependence of MCI on extracellular pH. A similar approach was used to study membrane partitioning of local anesthetics during VGSC inhibition[23,24,27].

We first examined memantine MCI with all extracellular solutions at pH 9.0 (Fig. 1c), a pH at which the [uncharged memantine] should be ~60-fold greater than at pH 7.2 (Fig. 1b). MCI of GluN1/2A receptors by 100 μM memantine was greatly augmented (Min $I_{MCI}/I_{Control}$ reduced) by raising the pH from 7.2 to 9.0 (Fig. 1c, d). In contrast, the traditional memantine $IC_{50}$ (measured with coapplication of drug and agonists) was ~2-fold higher at pH 9.0 ($IC_{50} = 3.43 \pm 0.61$ μM, $n = 5$) than pH 7.2 ($1.71 \pm 0.06$ μM, $n = 5$), consistent with previous blocker pH sensitivity measurements[38,39], suggesting that charged memantine mediates traditional channel block.

A limitation of the protocol used for Fig. 1c is that increasing pH might modify MCI by affecting membrane structure[40] or pH-sensitive NMDAR properties[41–43]. We therefore designed pH jump protocols in which NMDAR activation occurred at pH 7.2,

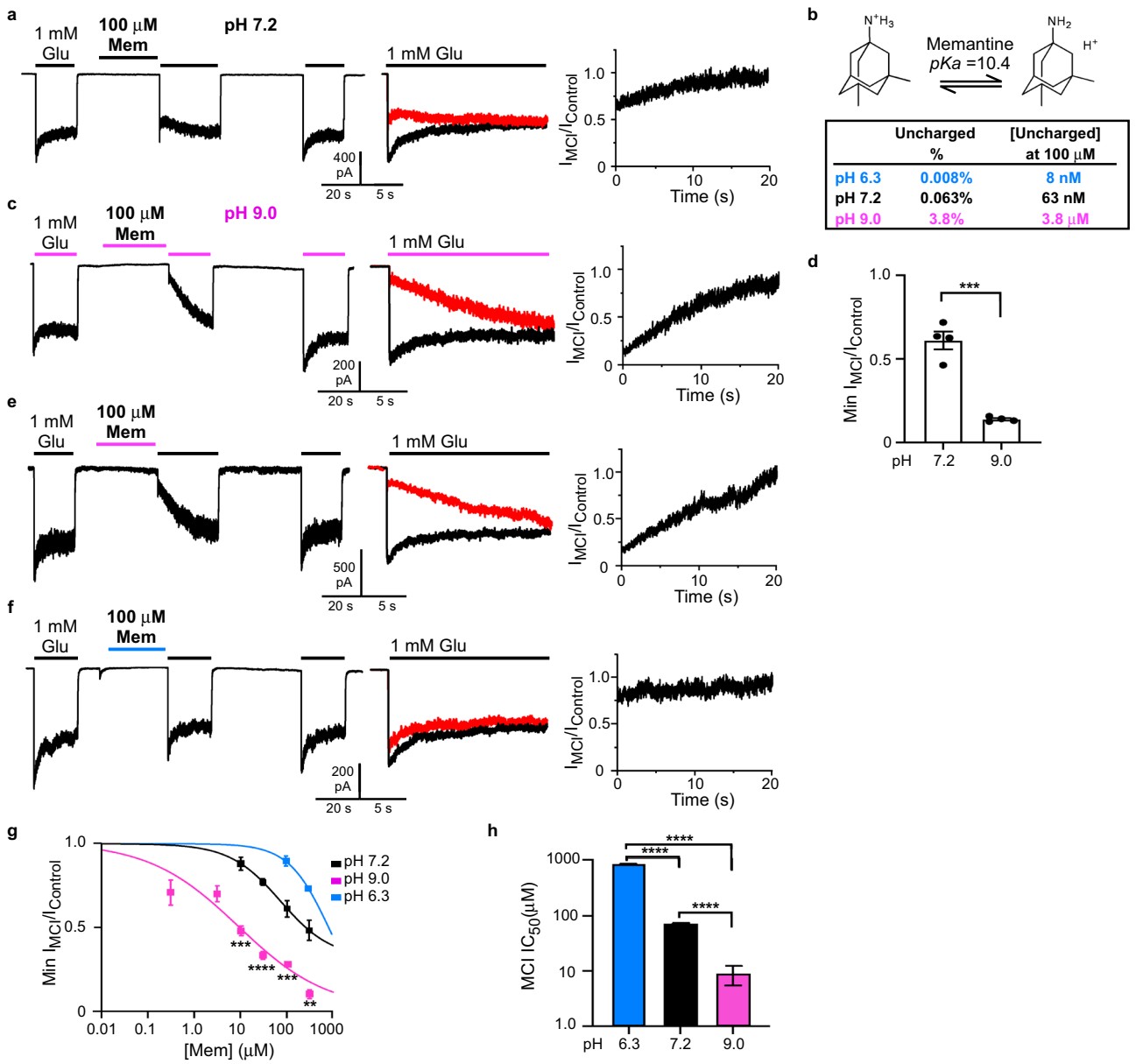

**Fig. 1 Memantine MCI is pH-dependent. a** MCI protocol at pH 7.2 with 100 μM memantine (left; Glu = glutamate; Mem = memantine); overlay of $I_{Control}$ and $I_{MCI}$ (middle; $I_{MCI}$ in red); point-by-point ratio, $I_{MCI}/I_{Control}$ (right). Similar current trace formatting is used in following figures. **b** Charged and uncharged memantine; amount uncharged at pH 6.3, 7.2, and 9.0. **c** MCI protocol with 100 μM memantine, pH of all solutions = 9.0. **d** Min $I_{MCI}/I_{control}$ with 100 μM memantine, pH of all solutions = 7.2 or 9.0 (pH 7.2, 0.611 ± 0.046, $n = 4$; pH 9.0, 0.139 ± 0.007, $n = 4$) compared with two-sided $t$-test ($t = 8.76$, df = 6, $p = 0.0001$). **e** pH 9.0 jump protocol with 100 μM memantine. **f** pH 6.3 jump protocol with 100 μM memantine. **g** [Memantine]-MCI curves at pH 6.3 (jump protocol), 7.2, and 9.0 (jump protocol). Only two memantine concentrations were used at pH 6.3 (see text). Min $I_{MCI}/I_{Control}$ values at 10 μM memantine (pH 7.2, $n = 4$; pH 9.0, $n = 4$) compared with two-sided $t$-test ($t = 6.94$, df = 6, $p = 0.0004$). Min $I_{MCI}/I_{Control}$ values at 30 μM memantine (pH 7.2, $n = 4$; pH 9.0, $n = 9$) compared with two-sided $t$-test ($t = 9.00$, df = 11, $p = 2.1*10^{-6}$). Min $I_{MCI}/I_{Control}$ values at 100 μM memantine (pH 6.0, $n = 5$; pH 7.2, $n = 4$; pH 9.0, $n = 4$) compared with one-way ANOVA (F(2, 10) = 65.1, $p = 1.9*10^{-6}$) and Tukey post-hoc test (pH 6.3 vs 7.2, $p = 0.00071$; pH 6.3 vs 9.0, $p = 1.3*10^{-6}$; pH 7.2 vs 9.0, $p = 0.00060$). Min $I_{MCI}/I_{Control}$ values at 300 μM memantine (pH 6.0, $n = 4$; pH 7.2, $n = 4$; pH 9.0, $n = 4$) compared with one-way ANOVA (F(2, 9) = 51.2, $p = 1.2*10^{-5}$) and Tukey post-hoc test (pH 6.3 vs 7.2, $p = 0.0082$; pH 6.3 vs 9.0, $p = 9.2*10^{-6}$; pH 7.2 vs 9.0, $p = 0.00048$). In **a**, **c**, **e**, **f**, color of lines above traces indicates solution pH (blue, pH 6.3; black, pH 7.2; magenta, pH 9.0; same color coding used in following figures). **h** Memantine MCI IC_{50}s (pH 6.3 jump, 841 ± 5 μM, $n = 5$; pH 7.2, 71.0 ± 1.7 μM, $n = 4$; pH 9.0 jump, 8.77 ± 3.26 μM, $n = 12$) compared with one-way ANOVA (F(2,18) = 1.23*10^{4}, $p < 1*10^{-15}$) and Tukey post-hoc test (pH 6.3 vs 7.2, $p = 2.7*10^{-14}$; pH 6.3 vs 9.0, $p = 2.7*10^{-14}$; pH 7.2 vs 9.0, $p = 1.0*10^{-8}$). All figures: $*p ≤ 0.05$; $**p ≤ 0.01$; $***p ≤ 0.001$; $****p ≤ 0.0001$. In **d**, **g**, **h**, mean ± SEM is plotted. In all figure legends, n is the number of biologically independent cells.

and pH was altered only during memantine application. The pH of the memantine-containing solution only was changed to pH 9.0 ("pH 9.0 jump") to increase, or pH 6.3 ("pH 6.3 jump") to decrease, the [uncharged memantine] (Fig. 1e–h). To reduce proton-activated currents, which are sensitive to high [memantine][44], pH jump experiments were performed in the continuous presence of 20 µM amiloride[45]. We calculated Min $I_{MCI}/I_{Control}$ for all pH jump experiments using a normalization procedure that controlled for possible delayed effect of the pH jump on NMDAR responses (Methods). Similar to MCI experiments performed entirely at pH 9.0, the pH 9.0 jump experiments showed that MCI by 100 µM memantine was greatly augmented (Min $I_{MCI}/I_{Control}$ reduced) relative to measurements at pH 7.2 (Fig. 1e). During pH 6.3 jump experiments, MCI by 100 µM memantine was greatly reduced relative to MCI at pH 7.2 (Fig. 1f).

To quantify carefully the pH dependence of MCI we estimated MCI $IC_{50}$s using [memantine]-MCI experiments at pH 7.2, with pH 6.3 jumps, and with pH 9.0 jumps (Fig. 1e–h). Note that at pH 6.3 we estimated MCI $IC_{50}$ using only two memantine concentrations (100 and 300 µM) because of low memantine MCI potency at pH 6.3, and only one parameter ($IC_{50}$) was free during fitting (see Methods). We found that memantine MCI $IC_{50}$ is powerfully pH dependent (memantine MCI $IC_{50}$ is ~100-fold greater at pH 6.3 than at pH 9.0; Fig. 1h), strongly supporting the hypothesis that uncharged memantine mediates MCI.

**A permanently charged memantine derivative does not exhibit MCI**. If MCI requires movement of uncharged memantine into the plasma membrane, then a permanently charged channel blocker should not exhibit MCI. To test this prediction we synthesized a memantine derivative with three methyl groups covalently attached to the nitrogen, creating the quaternary ammonium *N,N,N*,3,5-pentamethyladamantan-1-ammonium iodide (trimethyl memantine (TMM; Fig. 2a)). To compare the properties of TMM to a structurally similar memantine derivative with a titratable nitrogen, we also synthesized *N,N*,3,5-tetramethyladamantan-1-amine hydrochloride (dimethyl memantine (DMM, Fig. 2c)), which has two methyl groups attached to the nitrogen.

We first determined the potency of TMM and DMM as traditional channel blockers of GluN1/2A receptors. The traditional $IC_{50}$ of TMM was $72.3 \pm 14.2$ µM (Fig. 2b). The traditional $IC_{50}$ of DMM was $16.8 \pm 1.5$ µM (Fig. 2d), in reasonable agreement with a previous value ($28.4 \pm 1.4$ µM, measured using cultured neurons held at $-70$ mV)[46].

If MCI requires entry of uncharged molecules into the membrane, then TMM should not exhibit MCI at any pH. Predicting the pH dependence of DMM depends on its $pK_a$. Because there are no published estimates of DMM's $pK_a$, we calculated $pK_a$ values for memantine and DMM (Marvin 21.2, ChemAxon (https://www.chemaxon.com)), yielding a $pK_a$ of 10.7 for both memantine (in agreement with the measured $pK_a$ of 10.4 (ref. [36])) and DMM. Thus, the pH sensitivity of DMM MCI should be similar to that of memantine. We compared MCI by TMM and DMM at similar concentrations relative to their traditional $IC_{50}$s. Observing robust memantine MCI at pH 7.2, where the memantine MCI $IC_{50}$ is 71 µM (Fig. 1h), would require concentrations ~100-fold above memantine's traditional $IC_{50}$ (1.71 µM). Because of the high traditional $IC_{50}$s of TMM (72.3 µM) and DMM (16.8 µM) we were concerned that using 100-fold higher concentrations would lead to non-specific effects. We therefore performed experiments at constant pH 9.0, where memantine's MCI $IC_{50}$ is only moderately higher than its traditional $IC_{50}$. At pH 9.0, 1 mM TMM (~14 times its traditional $IC_{50}$ at pH 7.2) exhibited no MCI (Fig. 2e, g), whereas 165 µM

DMM (~10 times its traditional $IC_{50}$ at pH 7.2) displayed strong MCI (Fig. 2f, g). MCI by DMM and by 30 µM memantine (~16 times its traditional $IC_{50}$ at pH 7.2) were similar (Fig. 2g). Thus, permanently charged TMM does not exhibit MCI, whereas DMM, which differs from TMM by only one methyl group but (like memantine) has a titratable nitrogen, exhibits robust MCI. Figures 1 and 2 provide powerful support for the hypothesis that the uncharged forms of memantine and DMM mediate MCI.

**Numerous compounds exhibit MCI**. If the second site is the membrane rather than a true binding site, MCI should be exhibited relatively nonspecifically by NMDAR channel blockers with a titratable nitrogen. However, the path from membrane to channel may show some selectivity among channel blockers, as suggested by our previous observation that ketamine does not exhibit MCI[7] (see Discussion). Whether MCI is exhibited by other channel blockers has not previously been examined.

To determine whether MCI is widely expressed among NMDAR channel blockers with a titratable nitrogen, we tested the NMDAR channel blockers PCP (Fig. 3a, f), MK-801 (Fig. 3b, f), dextrorphan (Dex; Fig. 3c, f), and RL-208 (compound 8 in[47]; Fig. 3d, f). Each of these drugs displayed MCI, suggesting MCI is a broadly expressed mechanism of NMDAR channel blocker action. Drug concentrations that induced ~50% MCI based on preliminary experiments were chosen for Fig. 3. We also determined whether native NMDARs are subject to MCI by MK-801 (Fig. 3e). Inhibition by MK-801 was similar in recombinant GluN1/2A receptors and native NMDARs in cultured neurons (Fig. 3f). Native NMDARs also are subject to MCI by memantine[18]. Figure 3 data suggest that, for both GluN1/2A receptors and native NMDARs, MCI does not require blocker binding to a highly specific "second site".

**Blockers transit from a reservoir of drug molecules during MCI**. If the second site is the membrane, then MCI must result from transit of blocker from a "reservoir" of drug molecules within the membrane to the deep site. The principal alternative hypothesis (e.g.,[7]) is that the second site is an external binding site on NMDARs from which channel blockers transit to the deep site during MCI. We performed two types of experiments to distinguish these alternative hypotheses.

The first experiment involved measurement of the kinetics of drug transit from the second site to the deep site. If drugs transit from a reservoir to the deep site, then the kinetics of transit should depend on blocker concentration. In contrast, if the second site were a single true binding site, the kinetics of transit from second site to deep site would depend on the unidirectional transition rate, not on blocker concentration. To measure the kinetics of transit from second site to the deep site, we took advantage of a basic characteristic of the channel blockers examined here: they access the channel blocking site only after the channel opens. Thus, both the path through the extracellular gate and the path from membrane to channel must be occluded when the channel is closed. NMDAR current activated by the glutamate application after washout of blocker from the extracellular solution (Fig. 1a) therefore should reach an initial peak (the "preinhibition peak") before blocker can enter the open channel. After the preinhibition peak, current should decrease as blocker transits from membrane to channel.

Although a small preinhibition peak was often visible with memantine (e.g., Fig. 1a), the preinhibition peak current is typically smaller than control peak current, suggesting that memantine MCI starts before NMDAR current peaks. We hypothesized that the preinhibition peak is small because memantine MCI kinetics are relatively rapid. With 100 µM

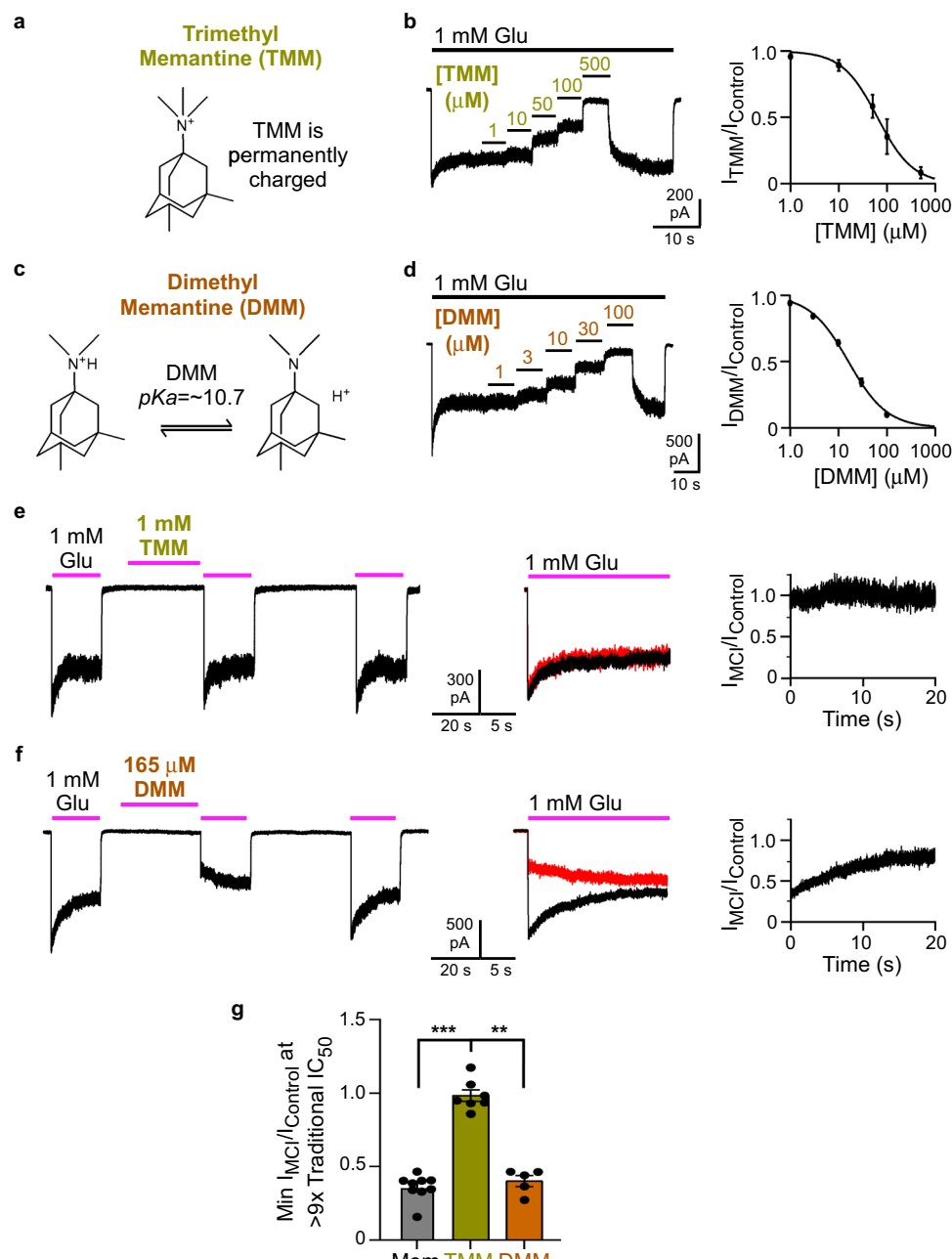

**Fig. 2 A permanently charged memantine derivative does not exhibit MCI. a** TMM, which exists in aqueous solution only in a charged form. **b** Recording protocol (left) used to measure traditional [TMM]-inhibition curve (right) at pH 7.2. TMM traditional $IC_{50} = 72.3 \pm 14.2$ μM, $n = 4$. **c** DMM, which exists in charged and uncharged forms in aqueous solution (pK$_a$ estimated computationally; see text). **d** Recording protocol (left) used to measure the traditional [DMM]-inhibition curve (right) at pH 7.2. DMM traditional $IC_{50} = 16.8 \pm 1.5$ μM, $n = 8$. **e** MCI protocol performed with 1 mM TMM at pH 9.0. **f** MCI protocol performed with 165 μM DMM at pH 9.0. In overlays in **e** and **f**, traces following TMM or DMM application are red. **g** Min $I_{MCI}/I_{Control}$ for 30 μM memantine ($0.327 \pm 0.024$, $n = 9$) in pH 9.0 jump experiments, and of 1 mM TMM ($0.987 \pm 0.039$, $n = 7$) and 165 μM DMM ($0.402 \pm 0.037$, $n = 5$) at constant pH 9.0. Min $I_{MCI}/I_{Control}$ values compared to 1 with two-tailed t-test. Values are different from 1 for memantine ($t = -11.0$, df = 3, $p = 0.0016$) and DMM ($t = -16.2$, df = 4, $p = 0.000086$), but not for TMM ($t = -0.339$, df = 6, $p = 0.75$). Min $I_{MCI}/I_{Control}$ values compared with one-way ANOVA ($F(2,18) = 16.1$, $p = 0.000099$) and Tukey post-hoc test (memantine vs TMM, $p = 0.00011$; memantine vs DMM, $p = 0.82$; TMM vs DMM, $p = 0.0019$). In **b**, **d**, **g**, mean ± SEM is plotted.

memantine at pH 7.2, the time constant of current decay following the preinhibition peak measured from point-by-point ratios was $46.4 \pm 4.1$ ms ($n = 9$). Because this measurement was made during simultaneous NMDAR activation and inhibition, the time constant of MCI onset is likely to be faster than 46.4 ms. Thus, memantine MCI onset is fast enough to partly inhibit NMDAR responses before current reaches its peak value, which,

with the perfusion system and other experimental conditions used here, typically requires 30–40 ms.

PCP, MK-801, Dex, and RL-208 exhibited pronounced pre-inhibition peaks and slow subsequent decays (Fig. 3a–d), suggesting these drugs transit from membrane to channel more slowly than memantine at the concentrations used. We used MK-801, the drug that appeared to exhibit the slowest kinetics of MCI onset, to

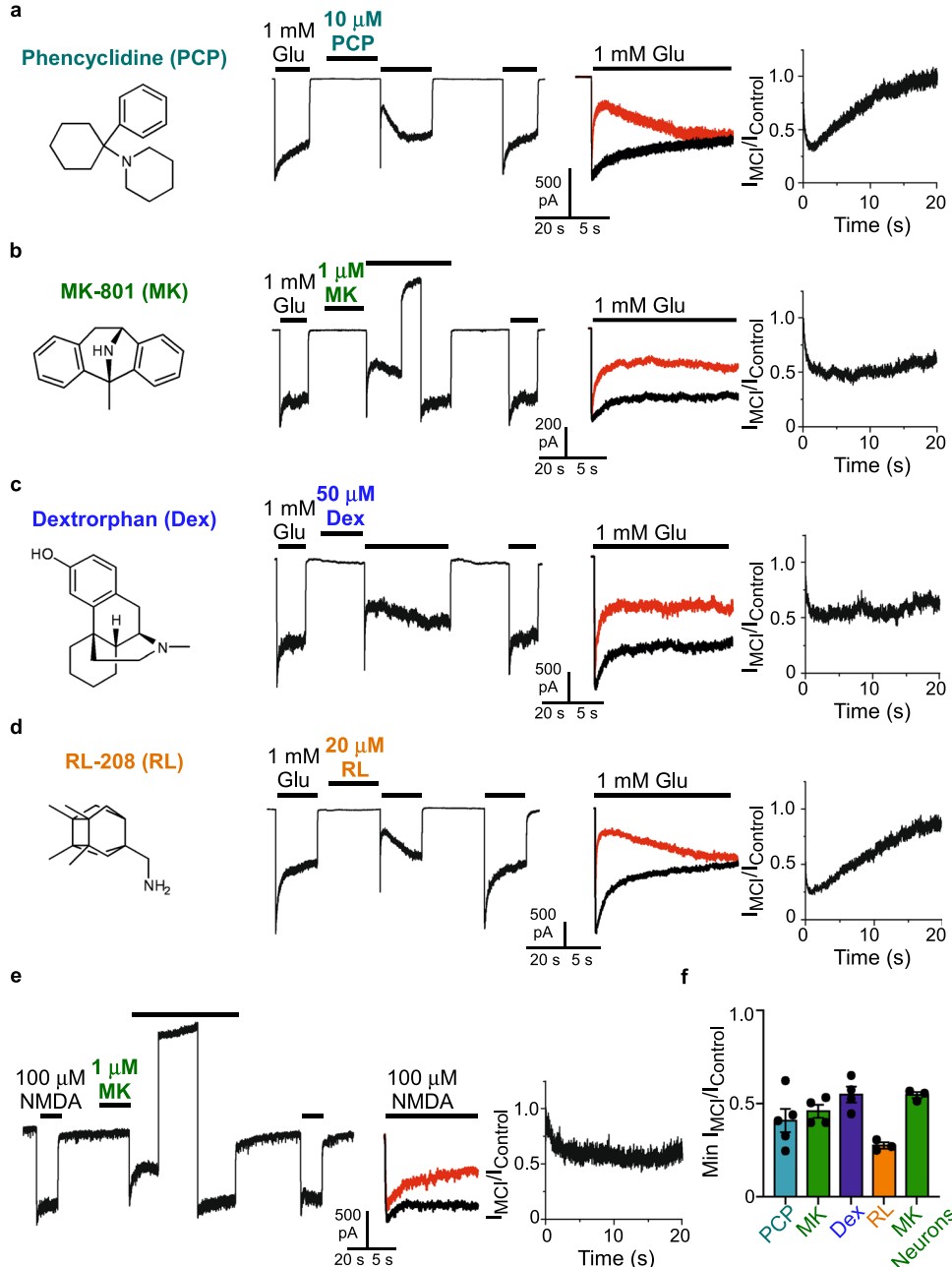

**Fig. 3 Multiple NMDAR channel blockers exhibit MCI. a–d** Chemical structures of uncharged forms of additional NMDAR channel blockers examined (left); examples of current traces during MCI protocols performed at pH 7.2 using GluN1/2A receptor-expressing tsA201 cells; overlays of $I_{Control}$ and $I_{MCI}$ ($I_{MCI}$ in red); $I_{MCI}/I_{Control}$ point-by-point ratios (right). Concentrations and blockers tested: 10 μM phencyclidine (PCP; **a**); 1 μM MK-801 (**b**), 50 μM dextrorphan (Dex; **c**), 20 μM RL-208 (**d**). **e** Examples of current trace during MCI protocol performed with 1 μM MK-801 at pH 7.2 using neurons in primary culture, overlay of $I_{Control}$ and $I_{MCI}$ ($I_{MCI}$ in red), and $I_{MCI}/I_{Control}$ point-by-point ratio (right). 50 μM APV was applied from 1 s before until 0.2 s after MK-801 application to ensure that channel openings during MK-801 application (which could allow traditional channel block) did not occur. All solutions used for neuronal recordings contained 1 μM tetrodotoxin and 1 μM Ro 25-6981; NMDARs were activated by application of 100 μM NMDA. In **b** and **e** a 20 s to 30 s $V_m$ step to 30 mV was performed after MK-801 application to speed unbinding from the deep site, allowing full recovery from MK-801 inhibition (needed because MK-801 has a much slower unbinding rate at −65 mV than the other blockers). **f** Min $I_{MCI}/I_{Control}$ based on the protocols shown in **a–e** for 10 μM PCP (0.409 ± 0.063, $n = 5$), 1 μM MK-801 (0.459 ± 0.035, $n = 4$), 50 μM dextrorphan (0.549 ± 0.043, $n = 4$), and 20 μM RL-208 (0.275 ± 0.017, $n = 3$) applied to tsA201 cells, and 1 μM MK-801 (0.612 ± 0.019, $n = 3$) applied to cultured neurons. Min $I_{MCI}/I_{Control}$ values were compared to 1 with two-sided t-test; all are significantly different from 1 (PCP, $t = −9.36$, $df = 4$, $p = 0.00073$; MK-801 with GluN1/2A receptors, $t = −15.5$, $df = 3$, $p = 0.00059$; Dex, $t = −10.5$, $df = 3$, $p = 0.0019$; RL-208, $t = −43.4$, $df = 2$, $p = 0.00053$; MK-801 with neurons, $t = −28.3$, $df = 2$, $p = 0.0013$). Mean ± SEM is plotted.

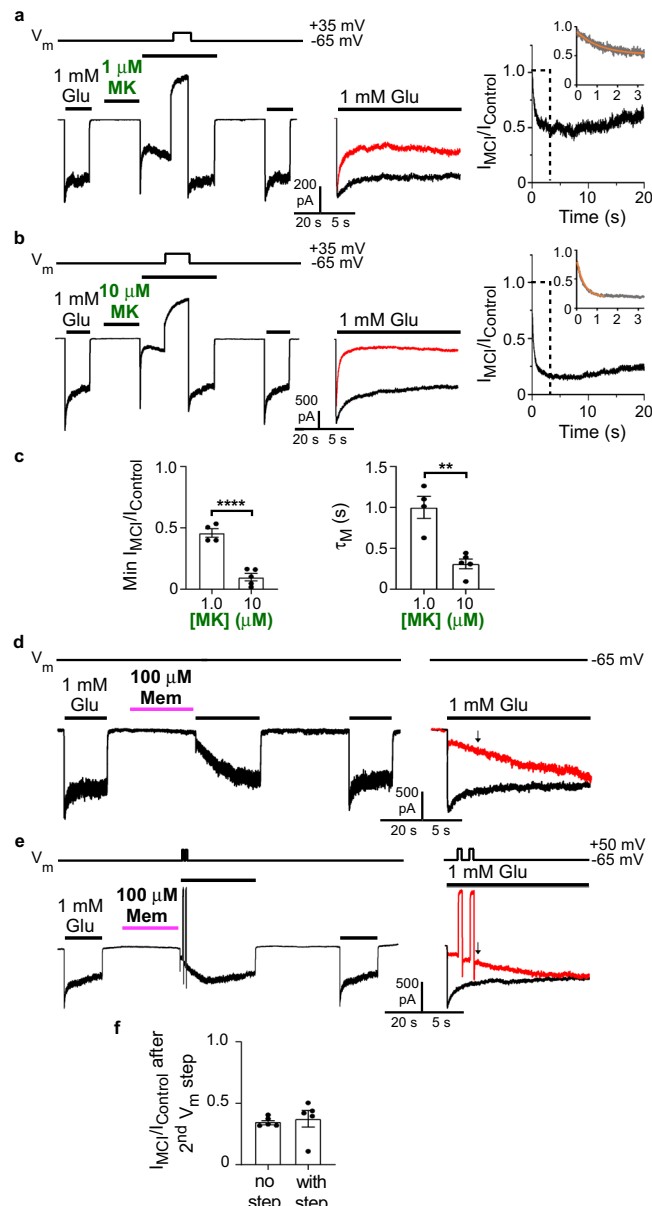

**Fig. 4 Blockers transit from a reservoir of drug molecules to the NMDAR channel during MCI. a–c** Measurement of [MK-801] dependence of $\tau_M$ at pH 7.2. **a, b** MCI protocol with 1 μM (**a**) and 10 μM (**b**) MK-801. A ~20 s $V_m$ step to 30 mV was performed after MK-801 application to allow full recovery from MK-801 inhibition. **c** Min $I_{MCI}/I_{Control}$ with 1 μM (0.459 ± 0.035; $n = 4$) and 10 μM MK-801 (0.097 ± 0.029; $n = 5$) compared with two-sided $t$-test ($t = 7.94$, df = 7, $p = 0.000096$; left); $\tau_M$ of MCI onset with 1 μM (1002 ± 135 ms, $n = 4$) and 10 μM MK-801 (311 ± 59 ms, $n = 5$) compared with two-sided $t$-test ($t = 5.06$, df = 7, $p = 0.0015$; right). **d, e** Investigation of memantine reblock during recovery from MCI. Memantine was applied using the pH 9.0 jump protocol because of the higher memantine potency observed at elevated pH (Fig. 1). Amiloride was used with cells that exhibited significant proton-activated currents. **d** Control (no $V_m$ jump) MCI protocol with 100 μM memantine (left), and overlays of $I_{Control}$ and $I_{MCI}$ ($I_{MCI}$ in red; right). **e** Same as **d**, except with two 500 ms $V_m$ steps to 50 mV performed during recovery from MCI. Arrows in the overlays in **d** and **e** show time of $I_{MCI}/I_{Control}$ measurement (time corresponding to 200 ms after the end of the second $V_m$ step). **f** $I_{MCI}/I_{Control}$ measured at the time shown by arrows in the overlays ($I_{MCI}/I_{Control}$: without $V_m$ steps (**d**), 0.350 ± 0.017, $n = 5$; with $V_m$ steps (**e**), 0.375 ± 0.069, $n = 5$) compared with two-sided $t$-test ($t = 0.344$, df = 8, $p = 0.74$). In **c**, **f**, mean ± SEM is plotted.

determine whether MCI onset is faster at a higher extracellular [MK-801] (resulting in a higher membrane [MK-801]).

We compared the time constant of MCI onset ($\tau_M$) using extracellular [MK-801]s of 1 and 10 μM. At both [MK-801]s we observed a clear preinhibition peak followed by a time-dependent current decay (Fig. 4a, b). The current decay was well fit by a single exponential (Fig. 4a, b, right, insets), allowing measurement of $\tau_M$, which was ~3-fold faster at 10 μM that at 1 μM MK-801 (Fig. 4c). The dependence of $\tau_M$ on [MK-801] is consistent with the hypothesis that during MCI, channel blockers transit to the deep site from a "reservoir" rather than from a single specific binding site.

The second experiment used to distinguish the drug reservoir and the specific binding site hypotheses took advantage of previous measurements of the voltage dependence of the MCI process. Although occupation of the second site is voltage-independent[8], inhibition after memantine transits to the deep site is voltage dependent[7,18]; as a result, MCI by 100 μM memantine is nearly abolished at 35 mV[8]. Therefore, we used two 500 ms $V_m$ steps from −65 mV to 50 mV during recovery from MCI (during the Glu application following removal of memantine) to induce

nearly complete unbinding of memantine from the deep site[48–51]. Note that this protocol would not be feasible with MK-801 because its slow unbinding kinetics would require much longer depolarizations. If the second site represents specific binding site(s) for one or a few molecules, then after the depolarizing steps there should be no memantine left to transit to the deep site. In this case, memantine reblock following the depolarizations should be slight or nonexistent. However, if the second site represents a memantine reservoir (and time course of recovery from memantine MCI represents reservoir depletion), then blocker should transit continuously from membrane to deep site during recovery from MCI. In this case, reblock should be observed following each depolarizing step. We performed MCI protocols without or with two $V_m$ steps imposed during recovery from MCI and measured $I_{MCI}/I_{Control}$ just after the end of the second $V_m$ step (Fig. 4d, e). $I_{MCI}/I_{Control}$ did not differ between experiments performed with or without the $V_m$ steps (Fig. 4f), suggesting that memantine can repeatedly bind to the deep site throughout MCI recovery. Thus, the "second site" cannot be a binding site for one or a few molecules per receptor (as assumed in[7,18–20]). A parsimonious explanation for these data is that the second site represents a reservoir of memantine in the plasma membrane.

**Modeling NMDAR open state fenestrations.** The above evidence indicates that MCI requires transit of channel blockers from the membrane to the deep site after NMDAR activation. Thus, there must be a path or fenestration in the NMDAR transmembrane domain (TMD) through which channel blockers can pass, but only when NMDAR channels are open. To attempt to identify such a path we developed two separate open state models of the NMDAR TMD (Model 1 and Model 2). For Model 1 we started with a previously developed closed GluN1/2A receptor TMD model[52]. We generated an open state structure using a steered molecular dynamics simulation protocol previously employed to model the AMPA receptor (AMPAR) TMD in the open state[53]. For Model 2 we started with a closed cryo-EM GluN1/2A NMDAR structure[54] and generated an open state structure using a homomeric GluA2 AMPAR structure in the open state[55] as a template. The NMDAR Model 2 open state structure in water and lipid is shown in Fig. 5a. Both our open state NMDAR models permitted flow of water through the

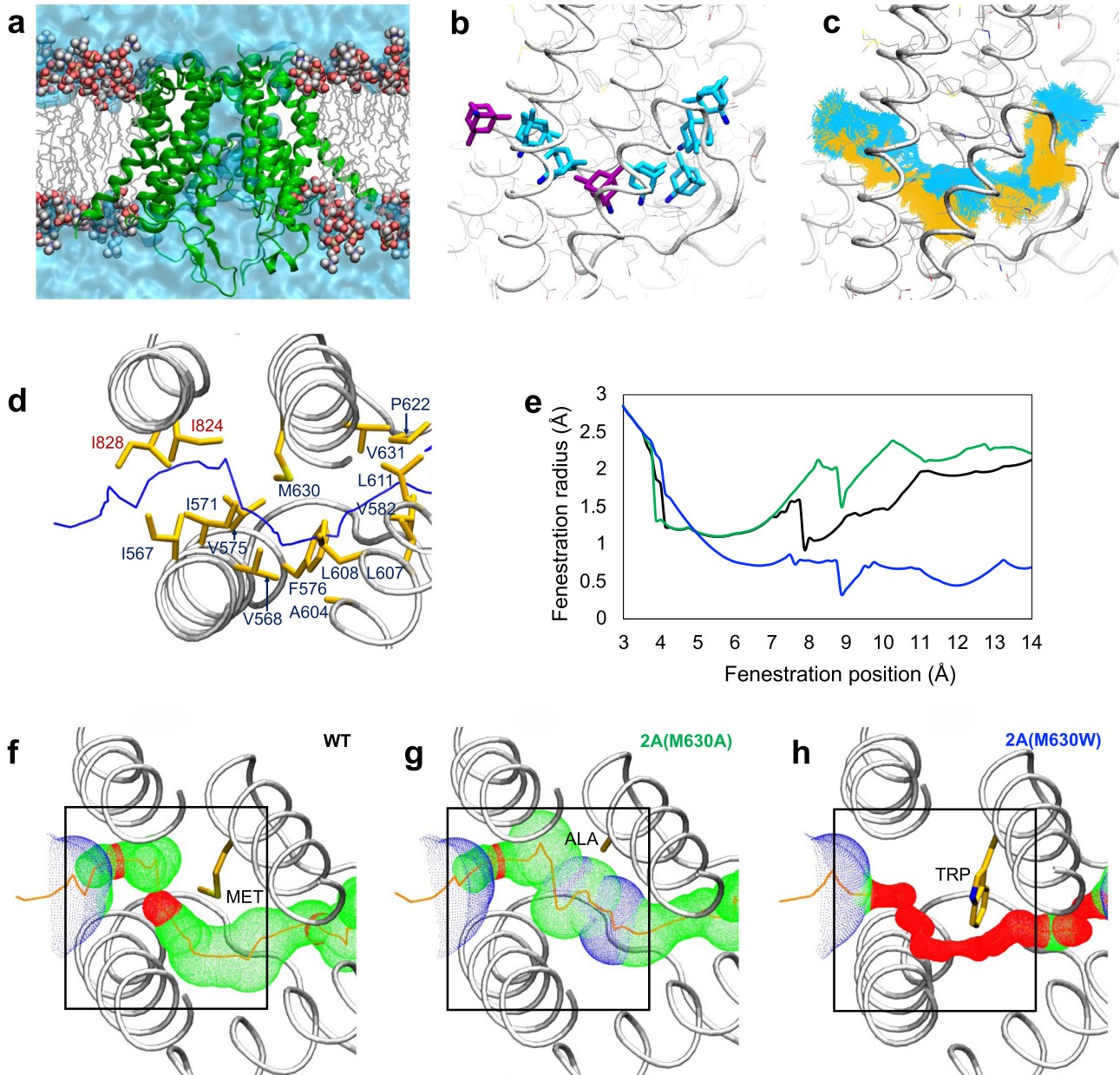

**Fig. 5 NMDAR TMD simulations reveal a state-dependent fenestration. a** Simulated open NMDAR TMD model (Model 2) in lipid bilayer and water. Protein is shown in green cartoon representation; lipid tails are grey wireframe; lipid head groups are spheres (carbon, grey; oxygen, red; phosphorus, brown; nitrogen, blue); water channel and bulk water are a solid blue surface. **b** Memantine (in stick representation) shown at multiple locations along the membrane-to-channel path. Snapshots of memantine are shown in the two docked positions (purple) used as starting points for steered MD simulations, and at multiple locations extracted from simulated trajectories (cyan). **c** Simulated trajectory of memantine from the membrane to the channel obtained using steered MD simulations with the following parameters: (1) $k = 10$ kcal mol$^{-1}$ Å$^{-2}$, $t = 60$ ns (yellow); and (2) $k = 4$ kcal mol$^{-1}$ Å$^{-2}$, $t = 100$ ns (cyan), where k = the biasing force constant and t = the total simulation duration. Simulation details are described in Methods. **b**, **c** were generated using Model 2. **d** View of the fenestration from the extracellular solution. The center of the fenestration path is shown as a blue line. Residues lining the channel are shown in yellow stick representation and identified (GluN1 residues, red labels; GluN2A residues, blue labels). **e** Fenestration radius along the path shown in **d** ("fenestration position") near position 630 of GluN2A is plotted for WT GluN1/2A (black line), GluN1/2A(M630A) (green line) and GluN1/2A(M630W) (blue line) receptors. Position 0 of the fenestration (not shown) corresponds to the outer edge of the NMDAR identified by HOLE. **f–h** The portion of the fenestration near position 630 of GluN2A is shown for WT GluN1/2A (**f**), GluN1/2A(M630A) (**g**), and GluN1/2A(M630W) (**h**) receptors. The largest regions of the fenestration identified using HOLE are shown in blue (radius > 2.30 Å), intermediate regions in green (1.15 Å < radius < 2.30 Å), and most constricted regions in red (radius < 1.15 Å). The residue at position 630 is identified and shown in stick representation. Black boxes show the portion of the fenestration plotted in **e**. **d–h** were generated using Model 1.

channel, and we observed diffusion of K$^+$ through the external gate of Model 2 (Supplementary Movie 1). However, it is important to acknowledge that these models may not represent fully open NMDAR states.

We identified multiple continuous paths from lipid to the ion channel in both closed and open NMDAR TMD structures using the pore predicting program HOLE[56]. We found a single lipid to channel path (fenestration) unique to the open structure. The

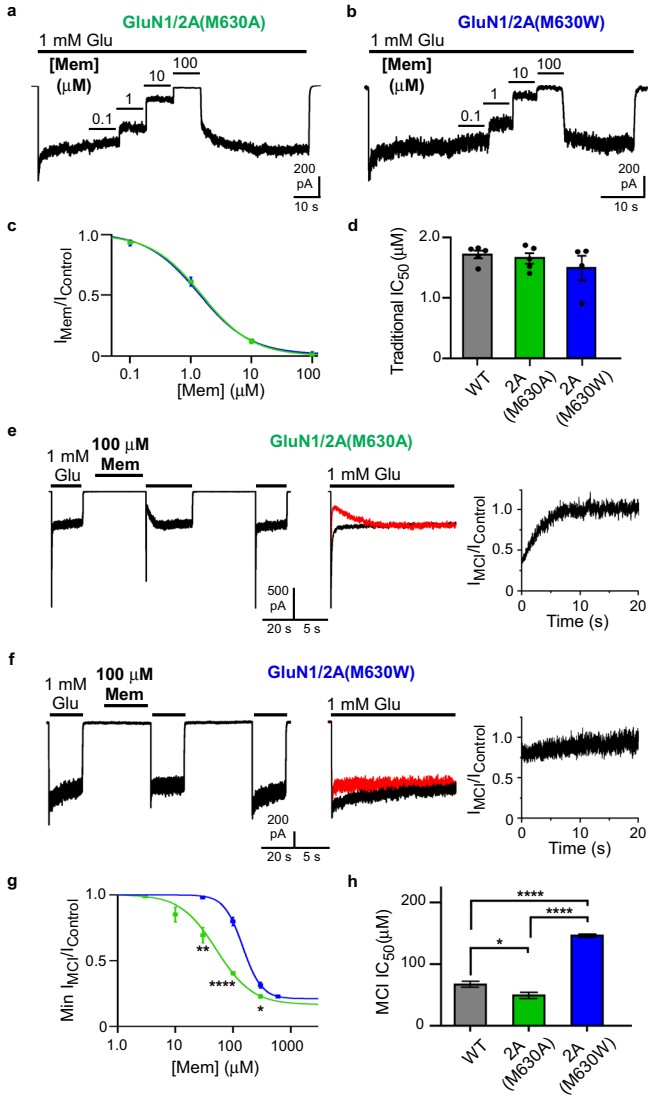

**Fig. 6 Mutation of predicted fenestration-lining residue GluN2A(M630) alters MCI. a, b** Recording protocols used to measure at pH 7.2 traditional [memantine]-inhibition curve for GluN1/2A(M630A) (**a**) and GluN1/2A(M630W) (**b**) receptors. **c** [Memantine]-inhibition curves used to measure traditional $IC_{50}$ for GluN1/2A(M630A) (green) and GluN1/2A(M630W) (blue) receptors. **d** Traditional $IC_{50}$s (WT, $1.71 \pm 0.06\,\mu M$, $n = 5$; GluN1/2A(M630A), $1.65 \pm 0.09\,\mu M$, $n = 5$; GluN1/2A(M630W), $1.49 \pm 0.20\,\mu M$, $n = 4$) were not significantly different based on one-way ANOVA ($F(2,11) = 0.855$, $p = 0.45$). **e, f** MCI protocol performed with $100\,\mu M$ memantine at pH 7.2 for GluN1/2A(M630A) (**e**) and GluN1/2A(M630W) (**f**) receptors. **g** [Memantine]-MCI curves used to measure MCI $IC_{50}$ for GluN1/2A(M630A) (green) and GluN1/2A(M630W) (blue) receptors. MCI $IC_{50}$ values at $30\,\mu M$ memantine (GluN1/2A(M630A), $n = 4$; GluN1/2A(M630W), $n = 3$) compared with two-sided t-test ($t = 4.07$, df $= 5$, $p = 0.0096$). MCI $IC_{50}$ values at $100\,\mu M$ memantine (GluN1/2A(M630A), $n = 7$; GluN1/2A(M630W), $n = 8$) compared with two-sided $t$-test ($t = 10.4$, df $= 13$, $p = 1.1*10^{-7}$). MCI $IC_{50}$ values at $300\,\mu M$ memantine (GluN1/2A(M630A), $n = 4$; GluN1/2A(M630W), $n = 3$) compared with two-sided $t$-test ($t = 3.37$, df $= 5$, $p = 0.020$). **h** Memantine MCI $IC_{50}$ of WT receptors ($68.6 \pm 4.8\,\mu M$; $n = 9$), GluN1/2A(M630A) receptors ($51.4 \pm 5.1\,\mu M$; $n = 11$), and GluN1/2A(M630W) receptors ($148 \pm 2.8\,\mu M$; $n = 12$) (right) compared with one-way ANOVA ($F(2,29) = 160$, $p < 1.0*10^{-15}$) and Tukey post-hoc test (WT vs GluN1/2A(M630A), $p = 0.025$; WT vs GluN1/2A(M630W), $p = 3.4*10^{-13}$; GluN1/2A(M630A) vs GluN1/2A(M630W), $p < 1.0*10^{-15}$). In **d**, **h**, mean $\pm$ SEM is plotted.

simulations of the structures. We found that increasing residue size with a GluN2A(M630W) mutation decreased fenestration radius, whereas decreasing residue size with a GluN2A(M630A) mutation increased fenestration radius (Fig. 5e–h and Supplementary Fig. 2). The mutated receptors were stable during 200 ns unrestrained Model 2 MD simulations (Cα RMSD < 2 Å with respect to the equilibrated wildtype (WT) channel), indicating that the mutation of residue GluN2A(M630) does not cause significant conformational changes to the protein backbone.

**Mutation of GluN2A(M630) specifically alters MCI.** We hypothesized that GluN2A mutations predicted to modify the fenestration radius would affect memantine transit from membrane to deep site, and thus would alter memantine MCI $IC_{50}$. To test this hypothesis experimentally we performed site-directed mutagenesis to create GluN2A(M630A) and GluN2A(M630W), mutations predicted by in silico mutagenesis (Fig. 5) to strongly affect fenestration diameter.

NMDAR TMD mutations could affect MCI either by modifying memantine transit from membrane to channel, or by modifying memantine binding to the deep site. To distinguish between these possibilities, we first compared the traditional memantine $IC_{50}$s of WT, GluN1/2A(M630A), and GluN1/2A(M630W) NMDARs by recording currents during coapplication of memantine and glutamate (to permit memantine access to the deep site from the extracellular solution; Fig. 6a, b) We found that neither GluN1/2A(M630) mutation affected the traditional memantine $IC_{50}$ (Fig. 6c, d).

We then compared the memantine MCI $IC_{50}$ of WT, GluN1/2A(M630A), and GluN1/2A(M630W) NMDARs. Mutation GluN2A(M630A) decreased memantine MCI $IC_{50}$ (increased potency; Fig. 6e, g, h), whereas mutation GluN2A(M630W) increased memantine MCI $IC_{50}$ (decreased potency; Fig. 6f–h). These results suggest that replacement of the fenestration-lining methionine with a smaller alanine facilitated memantine's passage through the fenestration; in contrast, replacement with the larger tryptophan partially occluded the fenestration. Thus, consistent with the predictions of in silico mutagenesis, the membrane-to-channel

path is formed by channel opening mainly as a result of repositioning of the hydrophobic side chains of residues in the M3 and M1 helices of the GluN2A subunits (Supplementary Fig. 1). We performed multiple-position molecular docking of memantine along the identified path in the open state models. In both open state NMDAR models, memantine docked near the path entrance at the outer edge of the protein and within the path close to the methionine residue GluN2A(M630) (Fig. 5b). Because Model 2 was the more stable open state NMDAR model in equilibrium molecular dynamics (MD) simulations, we used Model 2 to examine the ability of memantine to traverse the identified membrane-to-channel path. Using the two positions at which memantine docked as starting points, we performed steered MD simulations with a weak biasing force to facilitate movement of memantine along the fenestration. We obtained similar trajectories for memantine using two different biasing force constants (k) and simulation durations (t): $k = 10\,\text{kcal mol}^{-1}\,\text{Å}^{-2}$ and t = 60 ns; $k = 4\,\text{kcal mol}^{-1}\,\text{Å}^{-2}$ and t = 100 ns (Fig. 5c). Our results indicate that memantine can traverse the path shown in Fig. 5b, c.

Residues that line the fenestration are shown in Fig. 5d. We found that fenestration-lining residue GluN2A(M630) forms a constriction (Fig. 5d, e). To examine how GluN2A(M630) mutations influence the constricted region, we performed in silico mutagenesis followed by energy minimization and equilibrium MD

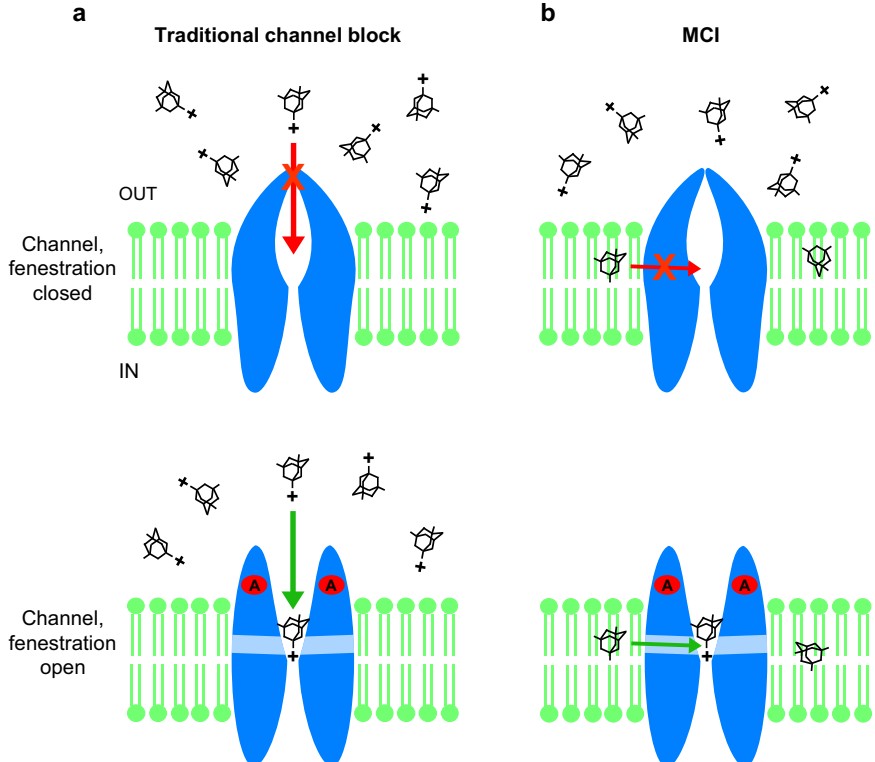

**Fig. 7 The two paths by which memantine can access the deep channel blocking site in NMDARs. a** Traditional channel block path. Top, charged memantine in the extracellular solution cannot access the deep site when the channel is closed. Bottom, charged memantine in the extracellular solution can transit to the deep site through the channel after agonists (A) bind and the channel gate opens. **b** MCI path. Top, uncharged memantine in the membrane cannot access the deep site when the membrane to channel fenestration is closed. Bottom, uncharged memantine in the membrane (shown here shortly after extracellular memantine has been washed away) can transit to the deep site through the gated fenestration after agonists bind and the fenestration opens.

path used by memantine during MCI is specifically altered in GluN1/2A(M630A) and GluN1/2A(M630W) receptors. These results demonstrate that there are two independent paths by which memantine can access the deep site and support the accuracy of the fenestration location predicted by our simulations.

**Mutation of additional fenestration lining residues**. We also examined the effect on memantine MCI of single or double mutations involving other residues predicted to line the fenestration (although, unlike GluN2A(M630), not at the narrow constriction): GluN1/2A(A570W), GluN1/2A(I571W), GluN1/2A(M630W)(A570W) and GluN1/2A(M630W)(I571W) (Supplementary Fig. 3). Cells transfected to express GluN1/2A(M630W)(I571W) receptors exhibited no glutamate-activated current. MCI by 100 μM memantine of GluN1/2A(I571W) and GluN1/N2A(M630W)(A570W) receptors did not differ significantly from the WT receptor value (Supplementary Fig. 3a). The Min $I_{MCI}/I_{Control}$ value was significantly lower for GluN1/2A(A570W) receptors than for WT receptors (Supplementary Fig. 3a). However, GluN1/2A(A570W) receptors also exhibited a lower traditional memantine $IC_{50}$ than WT receptors (Supplementary Fig. 3b). Thus, the increased MCI (decreased Min $I_{MCI}/I_{Control}$) exhibited by GluN1/2A(A570W) receptors could have resulted from increased memantine potency at the deep site rather than a specific effect on the MCI pathway.

**Discussion**
We investigate here MCI, a mechanism by which NMDAR channel blockers access their blocking site (the "deep site") and inhibit current. The two paths by which memantine, and other

NMDAR blockers, can access the NMDAR deep site are illustrated in Fig. 7.

NMDAR MCI occurs when drugs enter the plasma membrane and, after NMDAR activation, transit from the membrane to the deep site, where they block the channel. We focused here on GluN1/2A receptors, but also demonstrated MCI of native NMDARs. We first tested the hypothesis that MCI depends on uncharged memantine, an idea suggested by our previous observation[8] that occupation of the "second site" (from which memantine transits to the deep site during MCI) is voltage independent. We examined the dependence of MCI on pH because the concentration of uncharged (deprotonated) memantine increases as pH increases. Manipulation of pH has been used previously to modulate the protonation state of local anesthetics during inhibition of VGSCs[23,24,27]. We found that memantine MCI potency increased dramatically as pH was raised, even when pH was modified only during memantine application (Fig. 1). We then examined MCI by two memantine derivatives custom-designed and synthesized for this study (Fig. 2). Because of its quaternary ammonium, TMM is permanently charged. If MCI requires that uncharged channel blockers enter the membrane, TMM should not exhibit MCI. Similar use of a quaternary ammonium derivative was made to study fluoxetine inhibition of AMPARs[57]. DMM, in contrast, is a tertiary amine and exists in both charged and uncharged forms as a consequence of a pH-dependent equilibrium. We found that both TMM and DMM exhibit traditional channel block, whereas DMM, but not TMM, exhibits MCI. These data strongly support the conclusion that MCI depends on the uncharged form of memantine and its derivatives.

Uncharged memantine is strongly hydrophobic: the LogP of memantine is ~3[27,35–37], indicating that uncharged memantine is ~1000-fold more soluble in octanol than water. The vast majority of uncharged memantine thus resides in membranes, suggesting that MCI may begin with occupation of the plasma membrane by uncharged memantine. We tested several predictions based on the hypothesis that the second site is the membrane. First, since the uncharged form of any channel blocker should be able to enter the membrane, we examined whether MCI is exhibited by channel blockers other than memantine (and DMM). We found that the previously described channel blockers PCP, MK-801, dextrorphan, and RL-208 all exhibit MCI (Fig. 3). Thus, the second site appears non-specific, consistent with hypothesis that the plasma membrane is the second site. We also demonstrated that MK-801 exhibits MCI of native NMDARs in cultured cortical neurons (Fig. 3e). Interestingly, inhibition of neuronal NMDARs by very high intracellular [MK-801]s is a widely-used approach for inhibiting postsynaptic NMDARs (e.g.[58–62]). We propose that inhibition of neuronal NMDARs by high intracellular MK-801 is a consequence of MCI.

The channel blockers examined here exhibited MCI with less potency than traditional channel block, although the relation between traditional channel block potency and MCI potency varied. Thus, the preferred route of access of the blockers we examined may be from the extracellular solution, as would be expected, since these blockers were identified as traditional channel blockers. However, because the MCI protocol requires a 1-s wash before inhibition is quantified, we may have underestimated blocker MCI potency, especially if there is a rapid component of blocker exit from the membrane. Further studies may reveal that the preferred route of access of some of the blockers tested here, or of yet untested channel blockers, is through the membrane.

The NMDAR channel blocker ketamine is strongly lipophilic and has a molecular mass similar to that of PCP, but does not exhibit MCI[7,8]. This observation provides compelling evidence that our MCI protocol (Fig. 1) precluded access of channel blockers to the deep site from the extracellular solution (see Methods). It is unknown why ketamine does not exhibit MCI. Possible explanations include: (a) the membrane-to-channel path may be less permeable to ketamine than the other channel blockers tested; (b) ketamine may bind with high affinity to a site along the membrane-to-channel path, preventing access to the deep site; (c) after entering the plasma membrane, ketamine may exit so rapidly that membrane concentration approaches 0 during the 1 s wash used to eliminate blockers from the extracellular solution.

We tested two more predictions based on the hypothesis that the second site is the plasma membrane. First, the time course of MCI onset should depend on extracellular blocker concentration. If the second site is a specific binding site, the onset of MCI should depend only on the transition rate from the second site to the deep site. However, if the second site represents a reservoir of blocker molecules in the membrane, higher blocker concentrations in the extracellular solution (leading to higher concentrations in the membrane) should result in faster MCI onset. We used MK-801 for this test because the time constant of MCI onset ($\tau_M$), which for memantine is so fast that it could not be accurately resolved, is relatively slow for MK-801. We found a highly significant dependence of $\tau_M$ on [MK-801] (Fig. 4a–c). In addition, the preinhibition peak observed before MCI onset is consistent with the prediction that membrane-to-channel path must be gated, since the drugs used here are open channel blockers. Much smaller preinhibition peaks were observed with memantine, probably because of its very rapid transit from membrane to channel. Finally, we tested the prediction that, if the second site represents a reservoir of blocker molecules, then reblock from the reservoir should occur continuously during recovery from MCI.

When we used depolarizing steps to induce memantine unblock during recovery from MCI, we observed that the time course of recovery after repolarization was unaffected (Fig. 4d–f). This observation also is consistent with the dependence of MCI on a memantine reservoir, and furthermore suggests the time course of recovery from MCI is governed by memantine exit from the membrane.

An alternative to the MCI hypothesis is that uncharged channel blockers traverse the membrane and inhibit NMDAR responses by blocking at a site accessible from the intracellular solution. However, since channel blockers are too large to permeate the NMDAR channel[63], positively charged blockers that act from the intracellular solution should inhibit more effectively as $V_m$ is depolarized. In contrast, MCI is relieved by depolarization[8,18]. In addition, 30 μM intracellular memantine was found not to cause NMDAR inhibition[64]. MCI might be expected to be observed with blockers applied either intracellular or extracellularly, and it is possible inhibition would be observed with higher intracellular memantine concentrations. There is precedent for sidedness of inhibitor action: membrane-permeant VGSC blockers that can bind from the intracellular side of the membrane are ineffective when applied in the pipette solution during whole-cell recording[65].

We used structural molecular modeling to locate a membrane-to-channel path that memantine can transit only when the channel is open (a gated fenestration). We identified a residue (GluN2A(M630)) that forms a constriction in the fenestration observed in open channels (Fig. 5). In silico mutagenesis predicted that GluN2A(M630) mutations should modify fenestration radius. We tested model predictions by recording electrophysiologically from site-directed mutant NMDARs and found that GluN1/2A(M630W) and GluN1/2A(M630A) receptors displayed altered MCI without changes in traditional memantine IC$_{50}$ (Fig. 6). These results demonstrate that MCI and traditional channel block occur through independent paths. It is possible that GluN2A(M630) mutations alter transit of memantine from the membrane to the deep site through a mechanism other than direct disruption of the fenestration. However, the agreement of structural predictions and electrophysiological data provide strong support for the idea that GluN2A(M630) lines the fenestration. Closed NMDARs have been proposed to contain tunnels[33] that appear distinct from the fenestration identified here, through which lipids or small molecules may be able to access the receptor. We believe it is unlikely that fenestrations in closed NMDARs allow transit of open channel blockers, which can access and unbind from the deep site only when the NMDAR channel is open.

Receptor modulation of other ion channels through lipophilic pathways is well established. Local anesthetics can access their binding site in VGSCs via hydrophobic fenestrations[22,25,26,28], and permanently charged local anesthetics are unable to act on VGSCs through the hydrophobic path[22]. Pore access from the membrane through fenestrations also occurs in voltage-gated K$^+$ channels[66,67] and voltage-gated Ca$^{2+}$ channels[68–70]. Several iGluR ligands were previously proposed to act at hydrophobic sites with unknown properties, including: the NMDAR channel blockers MK-801[29] and ketamine[30]; the NMDAR inhibitor and local anesthetic bupivacaine;[32] cholesterol, which is required for NMDAR function;[31] the AMPAR inhibitor fluoxetine[57]. In addition, ligands for many other membrane proteins have been proposed to depend on partition into the plasma membrane, including antidepressants[71], cannabinoids[72], sphingosine 1-phosphate receptor ligands[73], and β2-adrenergic receptor agonists[74].

Thus, regulation of membrane proteins by ligands that travel though the plasma membrane is widespread. We expect that modulation of ionotropic glutamate receptors through

hydrophobic pathways, as exemplified by MCI of NMDARs, will be found to be a mechanism of broad significance.

## Methods

**Cell culture and transfection**. Experiments were performed on the tsA201 cell line (The European Collection of Authenticated Cell Cultures, Catalog No. 96121229), a variant of the HEK 293 cell line, and on primary cultures of rat cortical neurons. tsA201 cells were maintained[75] in DMEM supplemented with 10% fetal bovine serum and 1% GlutaMAX (Thermo Fisher Scientific). $1 \times 10^5$ cells/dish were plated on 15 mm glass coverslips treated with poly D-lysine (0.1 mg/ml) and rat-tail collagen (0.1 mg/ml, BD Biosciences) in 35 mm petri dishes.

Wildtype or mutant GluN1/2A receptors were used for all tsA201 cell experiments. 12–24 h after plating, tsA201 cells were transiently co-transfected using FuGENE6 Transfection Reagent (Promega) with mammalian expression plasmids that contained cDNAs encoding enhanced green fluorescent protein (EGFP in prK7) for identification of transfected cells, the rat GluN1-1a subunit (referred to as GluN1; GenBank X63255 in pcDNA3.1), and the rat GluN2A subunit (GenBank M91561 in pcDNA1). For some experiments cells were transfected with GluN1 and EGFP:pIRES:GluN2A (a generous gift from Dr. Kasper Hansen (Hansen, unpublished)), which was constructed by inserting EGFP in pIRES (Clontech) under transcriptional control of the CMV promoter, and rat GluN2A cDNA (GenBank D13211) after the IRES sequence. Voltage clamp recordings were performed on tsA201 cells 12–48 h after transfection.

Site-directed mutagenesis was performed on cDNAs encoding GluN1 and GluN2A subunit genes in ampicillin resistance-encoding plasmids (pcDNA 3.1 or pcDNA1) using the Stratagene Quik-Change XL sited-directed mutagenesis kit. Mutagenized NMDAR subunit cDNAs from isolated colonies were sequenced from 100–200 bases upstream to 100–200 bases downstream of each mutation (University of Pittsburgh Genomics and Proteomics Core Laboratories). cDNA ratios used in transfection were 1:1:1 (EGFP, GluN1, and GluN2A) or 1:1 (GluN1 and EGFP:pIRES:GluN2A). Following transfection, the competitive NMDAR antagonist D,L-2-amino-5-phosphonopentanoate (200 μM) was added to the culture medium to prevent NMDAR-mediated cell death.

Primary cultures of cortical rat neurons were prepared from day 16 Sprague-Dawley rat embryos[76] following procedures approved by the Institutional Animal Care and Use Committee of the University of Pittsburgh. Embryos were removed from pregnant rats sacrificed by $CO_2$ inhalation, embryonic cortical cells were dissociated using trypsin, and were plated on 12-mm glass coverslips (670,000 cells/well) in six-well plates. Cell proliferation was inhibited after 2 weeks with 1–2 μM cytosine arabinoside. Recordings were performed on cultured neurons between 19 and 25 days in vitro.

**Solutions**. The control extracellular bath solution contained (in mM): 140 NaCl, 2.8 KCl, 1 $CaCl_2$, 10 HEPES, 0.01 EDTA. pH was balanced to $7.2 \pm 0.05$ or $9.0 \pm 0.1$ with NaOH, or to $6.3 \pm 0.05$ with HCl. Osmolality was raised to $290 \pm 10$ mOsm with sucrose. For experiments with tsA201 cells, all extracellular solutions contained 0.1 mM of the NMDAR agonist glycine. For experiments with neurons in primary culture, all extracellular solutions contained 10 μM glycine, 1 μM tetrodotoxin to prevent action potential generation, and 1 μM Ro 25-6981 to inhibit GluN1/2B receptors. Intracellular (pipette) solution contained (in mM): 130 CsCl, 10 HEPES, 10 BAPTA, and 4 MgATP. pH was $7.2 \pm 0.05$ with CsOH. Osmolality was adjusted to $280 \pm 10$ mOsm.

Although pH 6.3 and 9.0 are outside the useful buffering range of HEPES ($pK_a$ ~7.5), we did not change pH buffer to avoid simultaneously changing two conditions (pH and pH buffer). To test how well pH was maintained in our experiments we prepared the pH 9.0 extracellular solution and left it at room temperature for 3 h (typical time from solution preparation to initiation of an experiment); the solution was loaded into the fast perfusion system reservoirs; solution was allowed to flow for 25 min (typical duration of experiments); a solution sample then was collected and its pH measured. The measured pH ($8.89 \pm 0.03$ ($n = 3$)) suggested that pH was adequately maintained in our experiments.

Drugs and their sources were: memantine (Tocris), trimethyl memantine and dimethyl memantine (see "Synthesis and purification of TMM and DMM" below), D-APV (Hello Bio), phencyclidine (Sigma-Aldrich), MK-801 (Hello Bio), dextromethorphan (Sigma-Aldrich), RL-208 (provided by SV; see[47]), NMDA (Tocris), tetrodotoxin (Abcam), Ro 25-6981 (Tocris), and amiloride hydrochloride (Tocris).

**Electrophysiology and fast perfusion**. Pipettes were pulled from borosilicate capillary tubing (Sutter Instruments) on a Flaming Brown P-97 microelectrode puller (Sutter Instruments) and polished with a heated filament to a resistance of 2–5 MΩ. Whole-cell recordings were made from cells expressing eGFP identified by epifluorescence illumination on an inverted Zeiss Axiovert microscope. Cells were held at a $V_m$ of −65 mV (corrected for a liquid junction potential of −6 mV) unless otherwise indicated. Whole-cell currents were recorded using an Axopatch 200B patch-clamp amplifier (Molecular Devices). Series resistance was compensated 80–90% using the prediction and correction circuitry. Currents were low-pass filtered at 5 kHz and sampled with a Digidata 1440 A at 10 or 20 kHz in pClamp10

(Molecular Devices). Current traces for presentation were refiltered offline in Clampfit 10.7 at 50 Hz. NMDAR responses were activated by fast perfusion of 1 mM glutamate (tsA201 cells) or 100 μM NMDA (cultured neurons).

Solutions were delivered to cells using a ten-barrel fast perfusion system[77]. A critical requirement of our fast perfusion system was that the 1 s wash between application of channel blocker and application of 1 mM glutamate (see Fig. 1a) effectively removed channel blocker from the extracellular solution. If channel blocker remained in the extracellular solution after the 1-s wash, measurements of MCI could have been contaminated by inhibition due to traditional open channel block. We are confident the 1 s wash fully exchanged the extracellular solution based on the following evidence: (1) the 1 s wash is >30-fold longer than the time constant of solution exchange (27 ms)[77]; (2) we demonstrated previously using our fast perfusion system that no NMDAR response inhibition was observed after: (a) a 1 s wash was used[8] to eliminate 50 μM D-APV (~200-fold above the D-APV $K_i$[78]); (b) after a 1 s wash was used[18] to eliminate 1.4 mM $Mg^{2+}$ (~250-fold above the $Mg^{2+}$ $IC_{50}$ at −65 mV[79]); (c) when a 0.4 s wash was used[7] to eliminate 500 μM ketamine (~500-fold above the ketamine traditional $IC_{50}$ at −65 mV[77]). The ketamine experiment provides a particularly stringent demonstration that our MCI measurements are not contaminated by inhibition due to traditional channel block.

The iodide salt of TMM was used here. To determine whether a compensatory effect of $I^-$ may have hidden MCI by TMM, we repeated the MCI protocol shown in Fig. 2e with 1 mM NaI (1 mM $Na^+ + 1$ mM $I^-$ in solution) replacing 1 mM TMM (1 mM $TMM^+ + 1$ mM $I^-$ in solution). We found no difference between Min $I_{MCI}/I_{Control}$ measurements with 1 mM NaI ($n = 4$) and with 1 mM TMM ($n = 7$; $t = 2.14$, df = 9, $p = 0.061$), suggesting that the presence of $I^-$ did not prevent observation of MCI by TMM.

**Analysis**. Data were analyzed with Clampfit 10.7 (Molecular Devices), Origin 16 or GraphPad Prism 7. Plots of $I_{MCI}/I_{Control}$ (plots to the right of current traces) were calculated by aligning the current traces to the time of 1 mM glutamate application and calculating the point-by-point ratio (similar to[80]) of $I_{MCI}$ divided by $I_{Control}$. $I_{Control}$ was calculated as the point-by-point average of $I_{Control1}$ (current activated by 1 mM glutamate before MCI) and $I_{Control2}$ (current activated by 1 mM glutamate after recovery from MCI).

The value of Min $I_{MCI}/I_{Control}$ was calculated as follows: (a) the minimum value of $I_{MCI}/I_{Control}$ was located and the mean $I_{MCI}/I_{Control}$ value over a 30 ms window centered on the minimum value was calculated; (b) the resulting value was normalized to the minimum value of control point-by-point ratios.

Normalization to the minimum value of control point-by-point ratios was performed because unnormalized minimum $I_{MCI}/I_{Control}$ values were biased to be <1 (even without inhibition) because we selected the minimum $I_{MCI}/I_{Control}$ value for measurement. In all experiments except pH jump experiments, the minimum value of control point-by-point ratios was calculated as the average of the minimum value (averaged over a 30 ms window) of the point-by-point ratios $I_{Control2}/I_{Control1}$ and $I_{Control1}/I_{Control2}$. We averaged minima of $I_{Control2}/I_{Control1}$ and $I_{Control1}/I_{Control2}$ to control for any possible rundown or runup of NMDAR response. For pH jump experiments, to control also for possible delayed effects of the pH jump on NMDAR responses, the minimum $I_{MCI}/I_{Control}$ value was measured using an additional set of experiments. In the additional control experiments, an identical pH jump protocol was performed except with [memantine] = 0; the procedure described above for determining minimum value of point-by-point ratios then was used. The minimum value of $I_{MCI}/I_{Control}$ in control (0 memantine) experiments used for normalization were: $0.906 \pm 0.031$ ($n = 6$) for the control pH 9.0 jump; $0.931 \pm 0.035$ ($n = 5$) for the control pH 6.3 jump.

To avoid inaccurate current quantification due to response run-down or run-up, MCI measurements were excluded if peak $I_{Control2}$ and peak $I_{Control1}$ differed by >20%. To minimize series resistance error, in addition to use of series resistance compensation cells were excluded from analysis if peak NMDAR current was >2.5 nA or if series resistance was >20 MΩ. Cells also were excluded from analysis if holding current was more negative than −200 pA (to avoid use of unhealthy cells) or if holding current fluctuations exceeded 100 pA during an experiment (to minimize inaccurate current quantification due to variation in holding current).

Memantine MCI $IC_{50}$ values for WT receptors at pH 7.2 and 9.0 (Fig. 1h) and for GluN1/2A(M630) mutant receptors (Fig. 6h) were calculated by fitting Eq. (1) to [memantine]-Min $I_{MCI}/I_{Control}$ curves (Figs. 1g and 6g).

$$\text{Min } I_{MCI}/I_{Control}([Mem]) = A + ((1 - A)/(1 + ([Mem]/IC_{50})^{nH})) \qquad (1)$$

where [Mem] is the concentration of memantine applied during the MCI protocol; Min $I_{MCI}/I_{Control}$ ([Mem]) is the value of Min $I_{MCI}/I_{Control}$ measured with [Mem]; $A$ is the value of Min $I_{MCI}/I_{Control}$ at saturating [Mem]; $IC_{50}$ is the memantine MCI $IC_{50}$; nH is the Hill coefficient. Equation (1) was fit to mean Min $I_{MCI}/I_{Control}$ values at each tested [memantine]. Free parameters during fitting were $A$, $IC_{50}$, and nH. We did not assume $A = 0$ because the 1 s wash between memantine application and glutamate reapplication during the MCI protocol (Fig. 1a) allowed some memantine to leave the membrane, potentially limiting maximal inhibition. Memantine MCI $IC_{50}$ value at pH 6.3 (Fig. 1h) was calculated using the same equation, but because only two memantine concentrations could be used we left only 1 parameter ($IC_{50}$) free. The value of $A$ was constrained to 0; nH was constrained to 1.

Traditional memantine $IC_{50}$s (Fig. 6d; Supplementary Fig. 3b), TMM $IC_{50}$s, and DMM $IC_{50}$s were calculated by fitting Eq. (2) to [drug]-inhibition curves (Fig. 2b, d; Fig. 6c).

$$I_{drug}/I_{Control}([drug]) = 1/(1 + ([drug]/IC_{50})^{nH}) \qquad (2)$$

where [drug] is the memantine, TMM, or DMM concentration at which fractional inhibition was measured; $I_{drug}/I_{Control}$ ([drug]) is fractional inhibition at [drug]; $IC_{50}$ is the memantine, TMM, or DMM $IC_{50}$; nH is the Hill coefficient. The equation was fit to $I_{drug}/I_{Control}$ values at all [drug]s from each cell. Free parameters during fitting were $IC_{50}$ and nH.

**Synthesis of *N,N,3,5-tetramethyladamantan-1-amine hydrochloride* (DMM).**
A solution of memantine (2.17 g, 12.1 mmol) in 15 ml of absolute EtOH was placed in a round-bottom flask equipped with a magnetic stirrer, a reflux condenser and an addition funnel. The solution was heated to 50 °C and formic acid (85% aq. sol., 2.75 ml, 50.7 mmol) was added slowly for 30 min. After that, formaldehyde (30% aq. sol., 4.56 ml, 45.4 mmol) was added dropwise with vigorous stirring for 1.5 h. When the addition of formaldehyde was completed, the reaction mixture was heated at 80 °C for 20 h. The resulting solution was then tempered to room temperature and the pH was adjusted to 12 with 5 M NaOH (10 ml). Dichloromethane (DCM) (30 ml) was then added, the phases were separated, and the aqueous phase was extracted with further DCM (2 × 10 ml). The combined organic phases were dried over anhydrous $Na_2SO_4$, filtered and concentrated *in vacuo* to yield an oily residue (1.88 g, 75% yield). Its hydrochloride was obtained by adding an excess of $HCl/Et_2O$ to a solution of the amine (885 mg, 4.27 mmol) in ethyl acetate, followed by filtration of the white precipitate (1.02 g, quantitative yield). The analytical sample was obtained as a white solid by crystallization from methanol/diethyl ether and contained less than 1 ppm of memantine as determined by HPLC/MS. Mp 180 °C. IR (ATR) ν: 3471, 3410, 2942, 2911, 2858, 2846, 2669, 2625, 2613, 2598, 2472, 1628, 1488, 1475, 1455, 1429, 1407, 1364, 1353, 1342, 1303, 1262, 1171, 1155, 1068, 1054, 1011, 998, 965, 935, 920, 894, 888 cm⁻¹. ¹H-NMR (400 MHz, $CD_3OD$) δ: 0.96 [s, 6H, 3(5)-$CH_3$], 1.22 (dt, J = 12.8 Hz, J' = 1.6 Hz, 1H, 4-$H_a$), 1.27 (dt, J = 12.8 Hz, J' = 1.6 Hz, 1H, 4-$H_b$), 1.37-1.47 [complex signal, 4H, 6(10)-$H_2$], 1.58 [dm, J = 12.0 Hz, 2H, 2(9)-$H_a$], 1.63 [dm, J = 12.0 Hz, 2H, 2(9)-$H_b$], 1.83 (m, 2H, 8-$H_2$), 2.33 (m, 1H, 7-H), 2.80 [s, 6H, N($CH_3$)_2]. ¹³C-NMR (100.6 MHz, $CD_3OD$) δ: 30.2 [$CH_2$, 3(5)-$CH_3$], 31.5 (CH, C7), 34.0 [C, C3(5)], 35.8 ($CH_2$, C8), 37.2 [$CH_3$, N($CH_3$)_2], 42.7 [$CH_2$, C6(10)], 43.1 [$CH_2$, C2(9)], 50.6 ($CH_2$, C4), 66.1 (C, C1). HRMS-ESI + m/z [M + H]⁺ calcd for $[C_{14}H_{26}N]^+$: 208.206, found: 208.206. Anal. Calcd for $C_{14}H_{25}N \cdot HCl \cdot H_2O$: C, 64.22; H, 10.78; N, 5.35. Found: C, 64.29; H, 10.61; N, 5.18.

**Synthesis of *N,N,N,3,5-pentamethyladamantan-1-ammonium iodide* (TMM).**
Methyl iodide (298 μL, δ = 2.28, 4.82 mmol) was added dropwise to a solution of *N,N,3,5-tetramethyladamantan-1-amine* (1.0 g, 4.82 mmol) in toluene (6 ml) at room temperature and stirred for 1 h. The white wax was filtered under vacuum and washed with toluene (10 ml) to yield a white solid (1.54 g, 91% yield). The analytical sample was obtained as a white solid by crystallization from methanol/diethyl ether and contained less than 1 ppm of memantine and less than 50 ppb of DMM as determined by HPLC/MS. Mp > 200 °C (dec.) (reported 290–293 °C)[81]. IR (ATR) ν: 3017, 2953, 2917, 2888, 2862, 2833, 1489, 1478, 1465, 1448, 1412, 1362, 1343, 1310, 1282, 1266, 1225, 1182, 1172, 1163, 1132, 991, 958, 947, 932, 920, 902, 842, 832, 799, 757. ¹H-NMR (400 MHz, $CD_3OD$) δ: 0.99 [s, 6H, 3(5)-$CH_3$], 1.21–1.29 (complex signal, 2H, 4-$H_2$), 1.39 [dm, J = 12.4 Hz, 2H, 6(10)-$H_a$], 1.47 [dm, J = 12.4 Hz, 2H, 6(10)-$H_b$], 1.39 [dm, J = 11.2 Hz, 2H, 2(9)-$H_a$], 1.47 [dm, J = 11.2 Hz, 2H, 2(9)-$H_b$], 1.98 (m, 2H, 8-$H_2$), 2.39 (m, 1H, 7-H), 3.05 [s, 9H, N($CH_3$)_3]. ¹³C-NMR (100.6 MHz, $CD_3OD$) δ: 30.3 [$CH_2$, 3(5)-$CH_3$], 32.2 (CH, C7), 34.3 ($CH_2$, C8), 34.8 [C, C3(5)], 41.6 [$CH_2$, C2(9)], 42.3 [$CH_2$, C6(10)], 49.05 ($CH_3$), 49.09 ($CH_3$) and 49.2 ($CH_3$) [N($CH_3$)_3], 50.2 ($CH_2$, C4), 75.7 (C, C1). HRMS-ESI + m/z [M + H]⁺ calcd for $[C_{15}H_{28}N]^+$: 222.2216, found: 222.2217. Anal. Calcd for $C_{15}H_{28}IN$: C, 51.58; H, 8.08; N, 4.01. Found: C, 51.58; H, 7.96; N, 3.89.

**Additional purification of DMM and TMM.** Because the traditional $IC_{50}$s of TMM and DMM are substantially higher than memantine's, DMM and TMM were extensively purified to avoid artifactual NMDAR inhibition by memantine contamination of DMM or TMM, or by DMM contamination of TMM. Products were recrystallized from methanol/diethyl ether twice and the purity of the samples was followed by HPLC/MS with limits of detection (LOD) of 1 ppm for memantine and 50 ppb for DMM. After the recrystallizations, DMM contained less than 1 ppm of memantine while TMM contained less than 1 ppm of memantine and less than 50 ppb of DMM.

See Supplementary Methods and Supplementary Figs. 4–11 in Supplementary Information for additional details on synthesis and purity of DMM and TMM.

**Modeling and molecular dynamics simulations.** All MD simulations were carried out using the pmemd.cuda program in the AMBER18 molecular dynamics package[82]. We used the FF12SB force field[83] (for open state NMDAR Model 1) or the FF14SB force field[84] (for open state NMDAR Model 2) for protein, the Lipid14 force field for lipids[85], the TIP3P water model, and the GAFF force field parameters for memantine (developed using the Antechamber module of AMBER). Channel

opening simulations for Model 1 were performed with a 1 fs integration step. For all other simulations an integration step of 2 fs was used and all covalent bonds to hydrogen atoms were constrained via SHAKE[86]. The Langevin thermostat and the Berendsen barostat[87] with anisotropic pressure scaling were used to maintain temperature and pressure. Long range electrostatic interactions were calculated using the Particle Mesh Ewald method with a cutoff radius of 10 Å. Periodic boundary conditions were applied in all directions. All simulations were performed with initial minimization of the systems using the steepest descent algorithm, followed by MD at 1 atmosphere pressure and 300 K temperature in constant-temperature, constant-pressure (NPT) ensemble.

To develop open state NMDAR Model 1, a MD-optimized closed channel model of the GluN1/2A NMDAR TMD in lipid bilayer and water was taken from our previous work[52]. The model contained residues 559 to 657 and 809 to 838 of GluN1 and residues 554 to 655 and 813 to 842 of GluN2A. The full simulated system contained 522 protein residues, 108 DMPC membrane lipid molecules, and 10159 water molecules and Na⁺ and Cl⁻ ions, resulting in a total of 43674 atoms. To obtain an open channel, a 10-ns steered MD simulation was performed (in 10 steps of ~1 ns each) with harmonic constraints applied to the SYTANLAAF sequence of all M3 helixes. The constraints were designed to gradually increase the distance between M3 helixes at the channel gate until the channel filled with water. To maintain structural integrity of the protein, backbone hydrogen bonds and dihedral angles of all TMD helices were harmonically restrained with a force constant of 20 kcal mol⁻¹Å⁻². The protocol was similar to the one used to produce an open AMPAR model[53].

To develop open state NMDAR Model 2, the cryo-EM structure of a closed state GluN1/2A NMDAR[54] (PDB ID: 6MM9 [https://www.rcsb.org/structure/6mm9]) was used as the starting template. The SYTANLAAF sequence of M3 helices of Model 2 was modeled based on the cryo-EM structure of an open AMPAR[55] (PDB ID: 5WEO [https://www.rcsb.org/structure/5WEO]). The open AMPAR structure contains an asymmetric external gate with kinked M3 helices in the B and D subunits, which correspond to the GluN2A subunits of our NMDAR model. Model 2 contained residues 550 to 657 and 809 to 838 of GluN1 and residues 545 to 655 and 813 to 837 of GluN2A. The model was placed in an equilibrated POPC lipid bilayer using CHARMM-GUI 3.2 Membrane Builder[88,89]. The full system in lipid bilayer and water contained 548 protein residues, 426 POPC membrane lipid molecules, 34413 water molecules and Na⁺ and Cl⁻ ions, resulting in a total of 169075 atoms. The open channel was equilibrated with gradually decreasing harmonic restraints on the protein backbone from 20 to 0.05 kcal mol⁻¹ Å⁻² over 100 ns followed by unrestrained MD simulations for 400 ns.

The program HOLE v2.2[56] was used to identify possible memantine paths from membrane to channel in representative structures of both open state NMDAR models. Multiple snapshots extracted from the equilibrium MD trajectory of Model 2 were analyzed with HOLE to select a structure with optimal side chain orientations for docking and subsequent MD simulations. Memantine (uncharged) was docked to the open state NMDAR models with AutoDock Vina 1.2.0[90,91] at 5 Å intervals along the path identified by HOLE using a grid box of 16 Å × 16 Å × 16 Å centered around path-lining residues. Each docked complex was energy-minimized and equilibrated for 10 ns with all protein Cα atom positions restrained with a force constant k = 1.0 kcal mol⁻¹ Å⁻². Steered MD simulations were carried out in two stages: (1) Memantine was pulled from the docked site at the path entrance into the protein by gradually decreasing the distance between the center of mass (COM) of memantine and Cα atoms of protein residues GluN2A(A604) and GluN2A(V631) from ~15 Å (initial value) to 0 Å. (2) Memantine was pulled from the docked site near GluN2A(M630) into the channel by gradually decreasing the distance between the COM of memantine and the COM of protein residues GluN1(M641) and GluN2A(V639) from ~20 Å until the ligand entered the ion channel. The Cα atom positions of helical segments of the protein were restrained using a force constant k = 40.0 kcal mol⁻¹ Å⁻² during both stages. The trajectories from the above two stages were combined to obtain the full path of memantine from the membrane to the channel. Two separate sets of "pulling simulations" were carried out with the following parameter values: (1) k = 10 kcal mol⁻¹ Å⁻² (k = 5 in AMBER) and t = 60 ns, and (2) k = 4 kcal mol⁻¹ Å⁻² (k = 2 in AMBER), t = 100 ns, where k = the biasing force constant and t = the total simulation duration. VMD 1.9.4[92] was used to visualize trajectories and generate molecular graphics.

**Statistics.** Statistical tests were performed in GraphPad Prism 7. The same sample was not measured repeatedly. We used one-way ANOVA with Tukey's *post hoc* analysis and two-tailed t-tests as indicated. For all electrophysiological experiments, n is the number of biologically independent cells. All error bars indicate ± standard error of the mean (SEM). Mean and SEM values for MCI $IC_{50}$s (Figs. 1h and 6h) were calculated by Origin 16 Nonlinear Curve Fitting of Min $I_{MCI}/I_{Control}$ mean ± SEM values (Figs. 1g and 6g).

**Reporting summary.** Further information on research design is available in the Nature Research Reporting Summary linked to this article.

## Data availability
All data that support the findings of this study are presented in this article, in Supplementary Information, and in the Source Data file. Additional information will be

made available from the corresponding author upon reasonable request. Previously published structures used in this study (PDB IDs 6MM9 [https://www.rcsb.org/structure/6mm9] and 5WEO [https://www.rcsb.org/structure/5WEO]) can be accessed at the Protein Data Bank (https://www.rcsb.org/). Source data are provided with this paper.

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

## Acknowledgements

We thank Elias Aizenman and Karen Hartnett-Scott for providing primary neuronal cultures, Kasper Hansen for providing the EGFP:pIRES:GluN2A construct, Kevin Jones for helpful comments on the manuscript, and Lihua Ming for excellent technical assistance. This work was supported by the US National Institute of Health grants R01GM128195 (JWJ) and R01AG065594 (JWJ). M.B.P was supported by US National Institute of Health grants T32NS007433 and F31NS113477. A.L.T. was supported by a fellowship (FPU grant) from the Spanish *Ministerio de Educación, Cultura y Deporte*.

## Author contributions

M.R.W., A.N., N.G.G., and M.B.P. acquired and analyzed data. M.R.W., N.G.G., A.N., M.B.P., C.N., M.G.K., and J.W.J. contributed to experimental design and data interpretation. C.N., S.M.-V. and D.P. performed molecular modeling and M.G.K. designed molecular modeling studies. A.L.T. and S.V. designed, synthesized, and purified memantine derivatives. All authors contributed to writing and revising the manuscript and approved its contents.

## Competing interests

The authors declare no competing interests.
