## [Peer Review File · Nature Communications]

Inhibition of NMDA Receptors Through a Membrane-to-Channel PathREVIEWER COMMENTS

Reviewer #1 (Remarks to the Author):

The study by Wilcox et al investigates the mechanism by which channel blockers inhibit NMDA receptors through a hydrophobic path in the transmembrane domain that allow the blockers to access the ion channel pore from the cell membrane. This previously undescribed "membrane-to-channel inhibition" (MCI) explains a number of elusive observations related to NMDA receptor channel block described in previous studies and Wilcox et al nicely interpret their results in the context of these previous observations. The study is timely considering recent years focus on the antidepressant effects of ketamine and highly relevant to the development of new channel blockers for clinical use. In fact, the study provides a compelling rationale to re-examine mechanisms of action of known channel blockers and paves the way for future studies on structural and functional features of MCI at NMDA receptors. Overall, the findings of this study are useful for future studies and are poised stimulate new experiments. The manuscript is well written, the experiments are elegantly constructed and carefully executed, the results are interpreted appropriately, and the figures are high quality. The conclusions drawn are reasonable and substantiated. I have no major comments or concerns, and only found a few minor points that might be worthwhile to address to improve the manuscript.

1) Lines 55-56: It would be helpful to include references that describe the examinations of therapeutic use for the channel blockers. In addition, amantadine is missing from the list.

2) Figure 1b: "mM" should be corrected to μM ".

3) Line 150: I paused a little to decode the meaning of "[memantine]s". It might be helpful to simply write out "memantine concentrations".

4) Line 219: Wouldn't some dependence on blocker concentration be expected if there are multiple second sites (e.g. 2 sites) on the receptor and the occupancy at this site depend on blocker concentration?

5) Line 235-237: Response 10-90% rise time using fast-application patch-clamp on lifted HEK cells is typically ~6-8 ms for GluN1/2A. It would therefore be useful to mention that the 30-40 ms rise time is for the perfusion system and experimental conditions of this study.

Reviewer #2 (Remarks to the Author):

The manuscript by Wilcox et al describes a unique mechanism of channel blockade in NMDA receptors, which they called "membrane-to-channel inhibition" (MCI). The authors implement electrophysiology and MD simulations in this study. The electrophysiology experiments are extensive and precise. Indeed, they are very nicely and carefully conducted. I have some concerns with the computational work insofar that the NMDAR model that has been generated through steered MD simulations does not in fact support the conclusions arising from the electrophysiology. Authors claim that MCI occurs out of a partial consequence resulting from a so-called 'gated fenestration' contained between NMDAR TMD helices.

>Despite this, the size of all ligands under investigation exceeds the size of the fenestration in the WT construct. Additionally, with the MCI potentiating mutant, M630A, larger compounds such as PCP also do not appear to be able to pass through this larger fenestration. The bulky tryptophan residue representing the M630W mutation could also undergo torsional changes that may open this fenestration, which notably was not checked by unbiased MD or at the very least inducible fit docking.

>Additionally, the authors claim that they have gone from a closed to open transition through the use of a steered MD simulation that occurs within the timeframe of 1 ns. This is an extremely abrupt conformational transition that it's likely to induce non-physiological changes to the underlying receptor structure. The use of a 1 femtosecond integration time-step further supports this notion as it should in theory be possible to integrate at 2 fs.

>Furthermore, authors claim that this is an 'open-state' structure, possibly because of the observation of water molecules in the channel pore. Without observing the permeation of ions, however, it might be a stretch to assign this model as definitely open. It may therefore be wise to tone down the language regarding the state assignment here. Using the method the authors described previously to generate an open state AMPAR (ref 48), they indeed show ion permeation. Could the authors do the same for NMDAR here?

>The authors also draw a conclusion about affinity with regards to distinguishing a binding site properly, from a pathway based on the results of docking free energy. Docking free energy scores are however notoriously unreliable and whilst I don't disagree with your assumption that this site outlines a pathway, I would be hesitant to say that on the basis of docking scores alone. Perhaps it would be better to just mention the size of the ligand/geometry of this pathway instead as putative descriptors of a pathway over a binding site.

> The authors describe in their methods section that the full simulated system contained 108 DMPC lipids. This really isn't very much and gives me concerns about possible minimum image convention violations. Could the authors please show that this has been accounted for?

Additional stylistic/ grammatical corrections:

Line 634: change 5N to 5M

Line 639: change precipitated to precipitate

Line 694: 'MD at room temperature,' please change to give temperature in kelvin.

Fig. 1b 3.8 mM -> 3.8 μ M

Reviewer #3 (Remarks to the Author):

This article describes a novel route by which the NMDAR blocker memantine reaches its site in the NMDAR channel's ion conduction path. The authors use electrophysiology, MD simulations and chemical synthesis to show that on activation memantine can reach its target by penetrating the plasma membrane and then laterally diffuse into the channel. The experiments are performed at a high standard, the paper is well written and of wider interest to readers in synaptic communication, ion channels and pharmacology.

Technically, it is a thorough study; particularly the e-phys experiments have been conducted with care, using a number of well thought-through controls, which are clearly described in the Methods. The results as presented appear solid and are supporting the findings well.

The authors also point out that a similar channel block has been shown for voltage activated channels. Hence, the reported pathway may be a more general channel blocking mechanism, thus contributing to our understanding of how drugs act on ligand gated receptors.

Other points:

Regarding the presentation of the MCI IC50 data at different pHs in Fig. 1 f&h: As described both in the text and Methods, the IC50 at pH 6.3 was determined using only two concentrations, 100 uM and 300 uM, giving an estimated IC50 of 855 uM. While the practical reasons for this are clear and the authors do not attempt to obscure the limitations of such an estimate in the text, showing these data in the Fig. 1h bar graph together with the (more appropriate) data sets for pH 7.2 and 9.0 does not seem fully justified, all the more so when the IC50 values were calculated using different equations (Methods). My suggestion would be to either omit the blue bar graph from Fig. 1h (and leave the data in text only) or add the data for pH 6.3 to Fig. 1f, which would make the limitations of this data set immediately obvious.

Fig. 1e legend appears to have a sentence that does not belong there (lines 951-954), describing some ANOVA parameters although Fig. 1e is only an example trace, with no statistics.

Also, the WT control data set appears to have exactly the same values as in Fig. 1h (MCI IC50 = 71.0 ± 1.7 uM; n = 4), suggesting the same set was re-used, could the authors comment/clarify.

The discussion of the computational studies is rather brief- more description and discussion would be very useful as there are certainly multiple interesting aspects that can be discussed from these. For example, the authors mention this pathway was the only one that did not appear in the closed state. Which residues/regions of the pathway help in keeping it closed?

The methods are also not very clear, e.g., if 10 steered MD simulations were done, did this result in 10 open models? If so, how was one chosen? Were the HOLE calculations then done on more than one model, and the proposed pathway appear consistently?

Fig. 5c shows the drugs docked at two points of the proposed pathway. However, there are constrictions between these points, how do the authors propose memantine to cross these barriers?

Reviewer #4 (Remarks to the Author):

This paper examines whether charged ammonium salts are able to act as NMDAR channel blockers via a what the authors describe as a 'hydrophobic path' distinct from the 'normal' route. In essence, the hypothesis is that only neutral (and hydrophobic) molecules can function via this (mostly non-specific) route, and that by modulation of charge (via N-alkylation or protonation state) this can be investigated.

The authors synthesize a couple of simple N-alkyl adamantylamine derivatives from the parent memantine via the classical Eschweiler-Clarke procedure (to give DMM whose charge state can be manipulated by pH) which is subsequently methylated with methyl iodide (to afford TMM, as a permanently charged derivative). The synthetic data for these materials look good; NMR spectra and other characterisation looks appropriate; purity appears high (and cross-contamination between these materials has been addressed).

The key experiment is that TMM can function as a 'traditional' GluN1/2A receptor channel blocker whereas it does not exhibit MCI (whereas DMM does in a pH sensitive manner). The challenge here is that the key observation is around a molecule (TMM) that does not actually elicit the (MCI) effect and directly linking this to the overall hypothesis (vs another unknown reason why TMM might not exhibit MCI). However, I think other reviewers may be better placed to evaluate this in the context of the rest of the paper; if they are supportive I would be content to see this in Nature Communications.

correction: line 363, I suggest: "DMM, in contrast, is a tertiary amine and exists in both charged and uncharged forms as consequence of a pH-dependant equilibrium."

POINT-BY-POINT RESPONSE TO THE REVIEWERS' COMMENTS ON NCOMMS-21-33662

Each reviewer comment is shown here verbatim, in *italics*, and is followed by the authors' response.

REVIEWER #1

REVIEWER COMMENT

The study by Wilcox et al investigates the mechanism by which channel blockers inhibit NMDA receptors through a hydrophobic path in the transmembrane domain that allow the blockers to access the ion channel pore from the cell membrane. This previously undescribed "membrane-to-channel inhibition" (MCI) explains a number of elusive observations related to NMDA receptor channel block described in previous studies and Wilcox et al nicely interpret their results in the context of these previous observations. The study is timely considering recent years focus on the antidepressant effects of ketamine and highly relevant to the development of new channel blockers for clinical use. In fact, the study provides a compelling rationale to re-examine mechanisms of action of known channel blockers and paves the way for future studies on structural and functional features of MCI at NMDA receptors. Overall, the findings of this study are useful for future studies and are poised stimulate new experiments. The manuscript is well written, the experiments are elegantly constructed and carefully executed, the results are interpreted appropriately, and the figures are high quality. The conclusions drawn are reasonable and substantiated. I have no major comments or concerns, and only found a few minor points that might be worthwhile to address to improve the manuscript.

AUTHORS' RESPONSE

We appreciate reviewer's insightful and supportive comments.

REVIEWER COMMENT

1) Lines 55-56: It would be helpful to include references that describe the examinations of therapeutic use for the channel blockers. In addition, amantadine is missing from the list.

AUTHORS' RESPONSE

Amantadine, and appropriate references, have been added (Introduction).

REVIEWER COMMENT

2) Figure 1b: "mM" should be corrected to μ M".

AUTHORS' RESPONSE

Thank you for catching that error; it has been corrected (Figure 1, panel b).

REVIEWER COMMENT

3) Line 150: I paused a little to decode the meaning of "[memantine]s". It might be helpful to simply write out "memantine concentrations".

AUTHORS' RESPONSE

"[memantine]s" has been changed to "memantine concentrations" (Results subsection "Memantine MCI depends on extracellular pH").

REVIEWER COMMENT

4) Line 219: Wouldn't some dependence on blocker concentration be expected if there are multiple second sites (e.g. 2 sites) on the receptor and the occupancy at this site depend on blocker concentration?

AUTHORS' RESPONSE

Yes, the reviewer is correct. We now state that the kinetics of transit to the deep site should be independent of blocker concentration only if there is a single second site (Results subsection "Blockers transit from a reservoir of drug molecules during MCI").

REVIEWER COMMENT

5) Line 235-237: Response 10-90% rise time using fast-application patch-clamp on lifted HEK cells is typically ~6-8 ms for GluN1/2A. It would therefore be useful to mention that the 30-40 ms rise time is for the perfusion system and experimental conditions of this study.

AUTHORS' RESPONSE

We revised the sentence to make clear that 30-40 ms is specific to our perfusion system and other experimental conditions (Results subsection "Blockers transit from a reservoir of drug molecules during MCI").

REVIEWER #2

REVIEWER COMMENT

The manuscript by Wilcox et al describes a unique mechanism of channel blockade in NMDA receptors, which they called "membrane-to-channel inhibition" (MCI). The authors implement electrophysiology and MD simulations in this study. The electrophysiology experiments are extensive and precise. Indeed, they are very nicely and carefully conducted. I have some concerns with the computational work insofar that the NMDAR model that has been generated through steered MD simulations does not in fact support the conclusions arising from the electrophysiology. Authors claim that MCI occurs out of a partial consequence resulting from a so-called 'gated fenestration' contained between NMDAR TMD helices.

AUTHORS' RESPONSE

We appreciate the reviewer's supportive comments on many aspects of the manuscript and address their concerns with the computational work below.

In the revised manuscript we base our conclusions on two separate open NMDAR models, the model used in the first manuscript submission (Model 1), and a second newly developed model (Model 2). Model 1 was developed before an experimental structure of an open ionotropic glutamate receptor channel was available. To help with our response to the reviewer's comments we developed Model 2 based on the cryo-EM structure of an open AMPA receptor (PDB: 5WEO, Twomey et al., 2017). The model was stable during a 400 ns unrestrained MD simulation. Analysis of the new open state model using the program HOLE revealed a membrane-to-channel path similar to the path we previously identified. In docking simulations, memantine docked to similar positions along the membrane-to-channel path. We carried out additional computational analysis using the new open NMDAR model; we updated the Results subsection "Modeling NMDAR open state fenestrations" and the Methods subsection "Modeling and molecular dynamics simulations"; we updated Figure 5; we new Supplementary Figures 1 and 2 to Supplementary Information; we added a new Supplementary Movie.

REVIEWER COMMENT

>Despite this, the size of all ligands under investigation exceeds the size of the fenestration in the WT construct. Additionally, with the MCI potentiating mutant, M630A, larger compounds such as PCP also do not appear to be able to pass through this larger fenestration. The bulky tryptophan residue representing the M630W mutation could also undergo torsional changes that may open this fenestration, which notably was not checked by unbiased MD or at the very least inducible fit docking.

AUTHORS' RESPONSE

We understand the reviewer's concern about whether the fenestration is large enough to accommodate channel blocking ligands. The constrictions in the path are formed by flexible side chains of residues lining the fenestration (residues GluN1(I824), GluN2A(I571), GluN2A(M630), GluN2A(L607) and GluN2(V631)). Because of side chain flexibility, the radius calculated by HOLE may not accurately represent whether a ligand is able to pass through the fenestration. To address

this issue, we performed steered MD simulations to pull a memantine molecule along the fenestration using a mild biasing force. We obtained a continuous membrane-to-channel path for memantine using a biasing force constant of $4 \text{ kcal mol}^{-1}\text{\AA}^{-2}$, indicating that memantine does fit through the proposed pathway.

Additionally, we carried out equilibrium MD simulations of GluN1/2A(M630A) and GluN1(M630W) receptors to explore the effect of side chain torsional changes. We found that, although the side chain of tryptophan in GluN1/2A(M630W) receptors does not completely block the fenestration, it forms a narrower constriction compared to the methionine in wild-type receptors (see Supplementary Fig. 2). Furthermore, the substitution of alanine for methionine in GluN1/2A(M630A) receptors opens up the path. These findings are consistent with our experimental results obtained using site-directed mutagenesis. We have updated the manuscript with these new results (Results subsection “Modeling NMDAR open state fenestrations”).

We have not yet examined, experimentally or computationally, how GluN2A(M630) mutations may affect transit of any of the larger channel blockers such as PCP from membrane to channel. We feel that addressing this interesting question is beyond the scope of the current manuscript.

REVIEWER COMMENT

>Additionally, the authors claim that they have gone from a closed to open transition through the use of a steered MD simulation that occurs within the timeframe of 1 ns. This is an extremely abrupt conformational transition that it's likely to induce non-physiological changes to the underlying receptor structure. The use of a 1 femtosecond integration time-step further supports this notion as it should in theory be possible to integrate at 2 fs.

AUTHORS' RESPONSE

The steered MD simulation used to open the channel for Model 1 was performed in 10 steps of ~ 1 ns, resulting in a total of 10 ns of simulation time. We apologize for the unclear wording in the previous version of the manuscript, and have revised the text for clarity. Please see the manuscript changes to the Methods subsection “Modeling and molecular dynamics simulations”, which include: “To obtain an open channel, a 10 ns steered MD simulation was performed (in 10 steps of ~ 1 ns each).”

Open state NMDAR Model 2, which was developed based on an open AMPAR structure, remained stable during 400 ns of unrestrained simulation with an integration time-step of 2 fs.

REVIEWER COMMENT

>Furthermore, authors claim that this is an ‘open-state’ structure, possibly because of the observation of water molecules in the channel pore. Without observing the permeation of ions, however, it might be a stretch to assign this model as definitely open. It may therefore be wise to tone down the language regarding the state assignment here. Using the method the authors described previously to generate an open state AMPAR (ref 48), they indeed show ion permeation. Could the authors do the same for NMDAR here?

AUTHORS' RESPONSE

We carried out an equilibrium MD simulation of the new open NMDAR model with increased K^+ concentration at the channel entrance and observed the permeation of a K^+ ion through the external gate (Supplementary Movie 1). However, we agree with the reviewer that, while our model represents an open channel conformation, it may not represent a fully open structure. We have revised our phrasing in the manuscript and added a disclaimer about the open channel model.

REVIEWER COMMENT

>The authors also draw a conclusion about affinity with regards to distinguishing a binding site properly, from a pathway based on the results of docking free energy. Docking free energy scores

are however notoriously unreliable and whilst I don't disagree with your assumption that this site outlines a pathway, I would be hesitant to say that on the basis of docking scores alone. Perhaps it would be better to just mention the size of the ligand/geometry of this pathway instead as putative descriptors of a pathway over a binding site.

AUTHORS' RESPONSE

We agree with the reviewer that docking scores are unreliable for predicting binding free energies, and it was not our intention to predict the binding affinity of the ligands based on docking scores. We have carried out a more thorough analysis with steered MD simulations by pulling a memantine molecule along the path using a weak biasing force (please see response to Reviewer #2, second comment). We have replaced the reference to docking free energies that appeared in the previously submitted manuscript with a discussion of the steered MD simulation results.

REVIEWER COMMENT

> The authors describe in their methods section that the full simulated system contained 108 DMPC lipids. This really isn't very much and gives me concerns about possible minimum image convention violations. Could the authors please show that this has been accounted for?

AUTHORS' RESPONSE

The reviewer is correct, the initially developed system was not designed for substantial MD simulations. The new simulated system contains 426 POPC lipids. The minimum distance between two periodic images of the protein is 36 Å.

REVIEWER COMMENT

Additional stylistic/ grammatical corrections:

Line 634: change 5N to 5M

AUTHORS' RESPONSE

"5N" was changed to "5 M" (Methods subsection "Synthesis of N,N,3,5-tetramethyladamantan-1-amine hydrochloride (DMM)").

REVIEWER COMMENT

Line 639: change precipitated to precipitate

AUTHORS' RESPONSE

"precipitated" was corrected to "precipitate" (Methods subsection "Synthesis of N,N,3,5-tetramethyladamantan-1-amine hydrochloride (DMM)").

REVIEWER COMMENT

Line 694: 'MD at room temperature,' please change to give temperature in kelvin.

AUTHORS' RESPONSE

We replaced 'room temperature' with '300 K' (Methods subsection "Modeling and molecular dynamics simulations").

REVIEWER COMMENT

Fig. 1b 3.8 mM -> 3.8 μM

AUTHORS' RESPONSE

Thank you for catching that error, which has been corrected (Figure 1, panel b).

REVIEWER #3

REVIEWER COMMENT

This article describes a novel route by which the NMDAR blocker memantine reaches its site in the NMDAR channel's ion conduction path. The authors use electrophysiology, MD simulations and chemical synthesis to show that on activation memantine can reach its target by penetrating

the plasma membrane and then laterally diffuse into the channel. The experiments are performed at a high standard, the paper is well written and of wider interest to readers in synaptic communication, ion channels and pharmacology.

Technically, it is a thorough study; particularly the e-phys experiments have been conducted with care, using a number of well thought-through controls, which are clearly described in the Methods. The results as presented appear solid and are supporting the findings well.

The authors also point out that a similar channel block has been shown for voltage activated channels. Hence, the reported pathway may be a more general channel blocking mechanism, thus contributing to our understanding of how drugs act on ligand gated receptors.

AUTHORS' RESPONSE

We appreciate the reviewer's careful description of the manuscript's implications.

REVIEWER COMMENT

Regarding the presentation of the MCI IC₅₀ data at different pHs in Fig. 1 f&h:

As described both in the text and Methods, the IC₅₀ at pH 6.3 was determined using only two concentrations, 100 μ M and 300 μ M, giving an estimated IC₅₀ of 855 μ M. While the practical reasons for this are clear and the authors do not attempt to obscure the limitations of such an estimate in the text, showing these data in the Fig. 1h bar graph together with the (more appropriate) data sets for pH 7.2 and 9.0 does not seem fully justified, all the more so when the IC₅₀ values were calculated using different equations (Methods). My suggestion would be to either omit the blue bar graph from Fig. 1h (and leave the data in text only) or add the data for pH 6.3 to Fig. 1f, which would make the limitations of this data set immediately obvious.

AUTHORS' RESPONSE

We agree it should be as clear as possible that the pH 6.3 data are more limited than, and were handled differently from, the pH 7.2 and pH 9.0 data. We appreciate that the reviewer suggested two alternatives. We decided to go with the second alternative, that is, to include the two pH 6.3 data points on the graph that now is shown in Fig. 1g (formerly Fig. 1f) along with a fit to the two data points. This required that we slightly modify how MCI IC₅₀ at pH 6.3 was calculated: we fit the equation described in the previously submitted version of the manuscript to the two pH 6.3 points, but left only a single parameter (IC₅₀) free during fitting (rather than calculating IC₅₀ twice from the two points and averaging the two values). We made clear in the text, in the Figure 1 legend, and in Methods that a different procedure was used for the pH 6.3 data than for the pH 7.3 and 9.0 data.

REVIEWER COMMENT

Fig. 1e legend appears to have a sentence that does not belong there (lines 951-954), describing some ANOVA parameters although Fig. 1e is only an example trace, with no statistics.

AUTHORS' RESPONSE

The confusing and poorly placed sentence included in the Fig. 1e legend was meant to describe statistical comparisons using the data shown in Fig. 1e. Because of the Fig. 1 modifications made in response to the previous comment from Reviewer 3, we believe this information now is provided more clearly.

REVIEWER COMMENT

Also, the WT control data set appears to have exactly the same values as in Fig. 1h (MCI IC₅₀ = 71.0 \pm 1.7 μ M; n = 4), suggesting the same set was re-used, could the authors comment/clarify.

AUTHORS' RESPONSE

The reviewer is correct that we used the same control data set ([memantine]-MCI curve at pH 7.2) in Fig. 1h and in Fig. 6F. To avoid possible concern with statistical comparisons, we

collected an additional pH 7.2 [memantine]-MCI curve that now is used for the comparison in Fig. 6F. The pH 7.2 [memantine]-MCI curve that originally was used both in Fig. 1h and in Fig. 6F now is used only in Fig. 1h.

REVIEWER COMMENT

The discussion of the computational studies is rather brief- more description and discussion would be very useful as there are certainly multiple interesting aspects that can be discussed from these. For example, the authors mention this pathway was the only one that did not appear in the closed state. Which residues/regions of the pathway help in keeping it closed?

AUTHORS' RESPONSE

We have updated the manuscript to include a more detailed discussion of the computational studies. We have also carried out additional MD simulations using a new open NMDAR model developed based on the cryo-EM structure of the open AMPA receptor (PDB: 5WEO, Twomey et al., 2017). We have updated the Results subsection "Modeling NMDAR open state fenestrations" and the Methods subsection "Modeling and molecular dynamics simulations".

The opening of the channel results in changes to the positions of M1, M3, and M4 helices. In both NMDAR models, distance between M3 and M1 helices increases slightly (roughly by ~1-1.5 Å at the fenestration) as the channel opens. This results in repositioning of several hydrophobic side chains lining the fenestration (see the new Supplementary Fig. 1 and associated text). However, we are not able to reliably predict conformational changes of the M4 helix using our NMDAR models. The position of the M4 helix varies since it is not covalently attached to the rest of the protein in either model.

REVIEWER COMMENT

The methods are also not very clear, e.g., if 10 steered MD simulations were done, did this result in 10 open models? If so, how was one chosen? Were the HOLE calculations then done on more than one model, and the proposed pathway appear consistently?

AUTHORS' RESPONSE

The channel opening MD simulation was performed on a single model in 10 steps of ~1 ns each, resulting in a total of 10 ns of simulation time. HOLE calculations were performed on a single representative structure of the open model to identify possible membrane to channel paths. We apologize for the unclear wording and have edited the manuscript to improve clarity. Please see changes to the Methods subsection "Modeling and molecular dynamics simulations", which includes: "To obtain an open channel, a 10 ns steered MD simulation was performed (in 10 steps of ~1 ns each)..."

REVIEWER COMMENT

Fig. 5c shows the drugs docked at two points of the proposed pathway. However, there are constrictions between these points, how do the authors propose memantine to cross these barriers?

AUTHORS' RESPONSE

The constrictions in the path are formed by flexible side chains of residues lining the fenestration (residues GluN1(I824), GluN2A(I571), GluN2A(M630), GluN2A(L607) and GluN2(V631)). We performed additional MD simulations using our new open NMDAR model to test if memantine can cross these barriers. Using the docked memantine positions as starting points, we carried out steered MD simulations to pull a memantine molecule along the fenestration using a weak biasing force. We obtained a continuous membrane-to-channel path for memantine using a biasing force constant of 4 kcal mol⁻¹Å⁻², indicating that memantine does fit through the proposed pathway. Please see changes to the Results subsection "Modeling NMDAR open state fenestrations".

REVIEWER #4

REVIEWER COMMENT

This paper examines whether charged ammonium salts are able to act as NMDAR channel blockers via a what the authors describe as a 'hydrophobic path' distinct from the 'normal' route. In essence, the hypothesis is that only neutral (and hydrophobic) molecules can function via this (mostly non-specific) route, and that by modulation of charge (via N-alkylation or protonation state) this can be investigated.

The authors synthesize a couple of simple N-alkyl adamantylamine derivatives from the parent memantine via the classical Escheiwer-Clarke procedure (to give DMM whose charge state can be manipulated by pH) which is subsequently methylated with methyl iodide (to afford TMM, as a permanently charged derivative). The synthetic data for these materials look good; NMR spectra and other characterisation looks appropriate; purity appears high (and cross-contamination between these materials has been addressed).

The key experiment is that TMM can function as a 'traditional' GluN1/2A receptor channel blocker whereas it does not exhibit MCI (whereas DMM does in a pH sensitive manner). The challenge here is that the key observation is around a molecule (TMM) that does not actually elicit the (MCI) effect and directly linking this to the overall hypothesis (vs another unknown reason why TMM might not exhibit MCI). However, I think other reviewers may be better placed to evaluate this in the context of the rest of the paper; if they are supportive I would be content to see this in Nature Communications.

AUTHORS' RESPONSE

We thank the reviewer for their positive assessment of the synthetic data and expression of flexibility. We agree that we cannot fully exclude the possibility that TMM does not exhibit MCI for a reason other than because it is permanently charged. However, we believe the experiments comparing MCI by DMM and TMM, in the context of the multiple additional experimental approaches used, provide strong and valuable support for our main conclusions.

REVIEWER COMMENT

correction: line 363, I suggest: "DMM, in contrast, is a tertiary amine and exists in both charged and uncharged forms as consequence of a pH-dependant equilibrium."

AUTHORS' RESPONSE

The recommended correction has been made (Discussion).

REVIEWERS' COMMENTS

Reviewer #1 (Remarks to the Author):

The authors have addressed all my points and further strengthened the study with additional experiments in response to concerns raised in the initial review. This is an elegantly executed study that provides important new understanding of NMDA receptor channel block.

Reviewer #2 (Remarks to the Author):

The authors addressed all of my major concerns. The manuscript is improved from the original version.

Reviewer #3 (Remarks to the Author):

We are happy with the revised paper, the authors have addressed all our points.

Reviewer #4 (Remarks to the Author):

I am pleased to see that the authors have made significant changes to the manuscript as advised by the reviewers and consequently I would be content to see this in Nature Communications.

POINT-BY-POINT RESPONSE TO THE REVIEWERS' COMMENTS ON NCOMMS-21-33662A

Each reviewer comment is shown here verbatim, in *italics*, and is followed by the authors' response.

REVIEWER #1

REVIEWER COMMENT

The authors have addressed all my points and further strengthened the study with additional experiments in response to concerns raised in the initial review. This is an elegantly executed study that provides important new understanding of NMDA receptor channel block.

AUTHORS' RESPONSE

We appreciate the reviewer's positive assessment of our revisions.

REVIEWER #2

REVIEWER COMMENT

The authors addressed all of my major concerns. The manuscript is improved from the original version.

AUTHORS' RESPONSE

We appreciate the reviewer's positive assessment of our revisions.

REVIEWER #3

REVIEWER COMMENT

We are happy with the revised paper, the authors have addressed all our points.

AUTHORS' RESPONSE

We appreciate the reviewer's positive assessment of our revisions.

REVIEWER #4

REVIEWER COMMENT

I am pleased to see that the authors have made significant changes to the manuscript as advised by the reviewers and consequently I would be content to see this in Nature Communications.

AUTHORS' RESPONSE

We appreciate the reviewer's positive assessment of our revisions.